# Prefrontal reinstatement of contextual task demand is predicted by separable hippocampal patterns

Jiefeng Jiang [1✉], Shao-Fang Wang[1], Wanjia Guo[2], Corey Fernandez[3] & Anthony D. Wagner [1,4]

Goal-directed behavior requires the representation of a task-set that defines the task-relevance of stimuli and guides stimulus-action mappings. Past experience provides one source of knowledge about likely task demands in the present, with learning enabling future predictions about anticipated demands. We examine whether spatial contexts serve to cue retrieval of associated task demands (e.g., context A and B probabilistically cue retrieval of task demands X and Y, respectively), and the role of the hippocampus and dorsolateral prefrontal cortex (dlPFC) in mediating such retrieval. Using 3D virtual environments, we induce context-task demand probabilistic associations and find that learned associations affect goal-directed behavior. Concurrent fMRI data reveal that, upon entering a context, differences between hippocampal representations of contexts (i.e., neural pattern separability) predict proactive retrieval of the probabilistically dominant associated task demand, which is reinstated in dlPFC. These findings reveal how hippocampal-prefrontal interactions support memory-guided cognitive control and adaptive behavior.

[1] Department of Psychology, Stanford University, Stanford, CA 94305, USA. [2] Psychology Department, University of Oregon, Eugene, OR 97401, USA.
[3] Neuroscience Program, Stanford University, Stanford, CA 94305, USA. [4] Wu Tsai Neurosciences Institute, Stanford University, Stanford, CA 94305, USA.
✉email: jiefeng.jiang@stanford.edu

Ahallmark of human intelligence is the ability to adaptively adjust behavior based on current task demands[1–4]. This ability depends on the representation of task-sets, which specify the task-relevance of stimuli, features and locations to attend to, and stimulus-action configurations that support the execution of a task through cognitive control[3,4]. Task-sets are known to be instantiated, in part, in neural activity patterns in lateral prefrontal cortex (PFC)[5–13]. As goals and environments change, it is necessary to retrieve new task-sets to replace ones that are no longer relevant. While the currently relevant task-set is sometimes explicitly cued, such as when a supervisor directs an employee to perform a specific task, in other situations, knowledge about the likely relevant task-set comes from past experience and can be reinstated via mnemonic mechanisms. One source of knowledge about the likely relevant task-set is associations between the spatial context in which goal-directed behavior is expressed and past task demands. In real life, spatial contexts are often strong predictors of the likely tasks that will be performed in the contexts (e.g., kitchen-cooking, library-reading). As such, learned associations between spatial contexts and demands of the to-be-performed tasks (hereafter, *contextual task demand*, CTD) may permit proactive control, wherein a subsequent encounter with a specific context triggers the retrieval (reinstatement) of the most strongly associated task demand.

A wealth of behavioral, computational, and neuroimaging data inform theories of proactive control, yet fundamental questions remain regarding how the brain acquires knowledge of and proactively retrieves probabilistically likely task-sets. Notably, task-sets are different from other knowledge about "what" (e.g., sensory information, semantic knowledge), in that task-sets include additional instructions on "how" to perform the tasks (e.g., how information should be processed, how attention should be directed, and how evidence maps to action selection). This difference warrants separate investigations of the mnemonic mechanisms of task-sets. On the acquisition end, previous studies have shown that neural structures implicated in probabilistic learning, such as the striatum, support the learning of cognitive control demand[14–16]. However, the neurocognitive mechanisms enabling the proactive retrieval of CTD remain unclear.

Given recent evidence that the hippocampus contributes to the retrieval of associative memories[17], guiding spatial attention[18,19] and adjusting cognitive control[20], we hypothesize that the retrieval of CTD depends on the hippocampus. Based on our hypothesized neurocognitive process model in Fig. 1, we predict that entering a spatial context will trigger the hippocampus to retrieve its associated CTD, which is reinstated in the PFC and influences goal-directed behavior. To test these predictions, we first assess the reinstatement of CTD in the PFC and its behavioral relevance. Next, we examine the link between CTD reinstatement in PFC and its relationship with hippocampal mnemonic mechanisms––in particular, pattern separation, which refers to the specialized hippocampal mechanism of assigning distinct hippocampal coding for similar items or events, which serves to reduce interference and facilitate discrimination among memory traces[21,22].

Following this logic, we develop a behavioral experiment leveraging 3D virtual environments to induce associations between four spatial contexts and two distinct task demands (two contexts per task demand). Using this paradigm and reinforcement learning models, we find behavioral evidence of the learning of CTD. In concurrently acquired fMRI data, we observe brain activity patterns reflecting the reinstatement of CTD in the right dorsolateral (dl)PFC when the participants entered the spatial context and prior to the start of the task. Further, CTD reinstatement is correlated with the similarity between hippocampal activity patterns, suggesting that retrieval of task demand

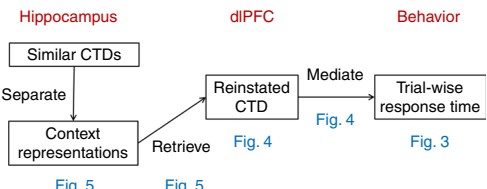

**Fig. 1 Proposed neurocognitive processes underlying the reinstatement of learned CTD.** Blue text indicates the figures where relevant results are shown.

representations relate to the representational similarity of hippocampal context codes.

## Results

**Behavioral results.** The experiment consisted of 6 runs of 8 blocks each, during which participants needed to draw on selective attention to make perceptual decisions about compound stimuli (an overlapping face/object pair). In each block, participants ($n = 33$) were cued to navigate to one of four buildings in a 3D environment. After entering each building, its interior was shown for 7.75 s, followed by eight perceptual decision-making trials (Fig. 2a). Each trial started with a task cue, followed by a face and object image pair. Based on the task cue, participants indicated either the gender of the face (male/female) or the type of object (clothes/tool) via a button press (Fig. 2). To manipulate CTD, participants performed 75% face/25% object trials in two contexts/buildings and 75% object/25% face trials in the other two contexts/buildings (Fig. 2c; see the "Methods" section for details).

Participants achieved high accuracy on the face (mean ± SEM: 0.89 ± 0.02) and object (0.90 ± 0.01) tasks, but were slower on object (1158 ± 38 ms) than on face (1094 ± 37 ms) trials ($t_{32} = 5.37$; uncorrected $P < 0.00001$, paired $t$-test; $d = 0.93$). A seemingly straightforward way to test the learning of CTD would be to compare the behavioral performance between conditions when the required task was congruent with the CTD (e.g., face task in a context of 75% face trials) to when it was incongruent (e.g., face task in a context of 75% object trials). However, this test can be confounded by other learning strategies that (partially) capture the statistical contingency. For example, one can in theory employ a temporal learning strategy, which makes predictions based on previous trials and ignores transitions of contexts, to achieve accurate prediction in most trials except for when there is a change of CTD between blocks. To control for temporal learning and to determine whether participants learned the task demand probabilistically associated with each context (i.e., the CTD), we performed reinforcement learning model-based, trial-level analyses (Fig. 3a). One reinforcement learning model (contextual model) simulated the learning of CTD[16], and a second reinforcement learning model (temporal model) simulated context-insensitive learning of task demand through temporal information[23,24]. Prior work indicates that model-based predictions of task demand facilitate behavioral performance when the predictions match the actual task demand[23,24]. In an analogous manner, we tested whether the CTD was learned by (a) computing trial-level prediction error in the contextual model, defined as the discrepancy between the predicted and actual task demand, and (b) determining whether contextual prediction error accounted for variance in trial-wise accuracy and response time (RT). Prediction error from the temporal model and the task required were used as covariates (see the "Methods" section: "Behavioral analysis").

Analyses revealed that the temporal model prediction error scaled with RT (regression coefficient: 0.03 ± 0.004; $t_{32} = 9.05$;

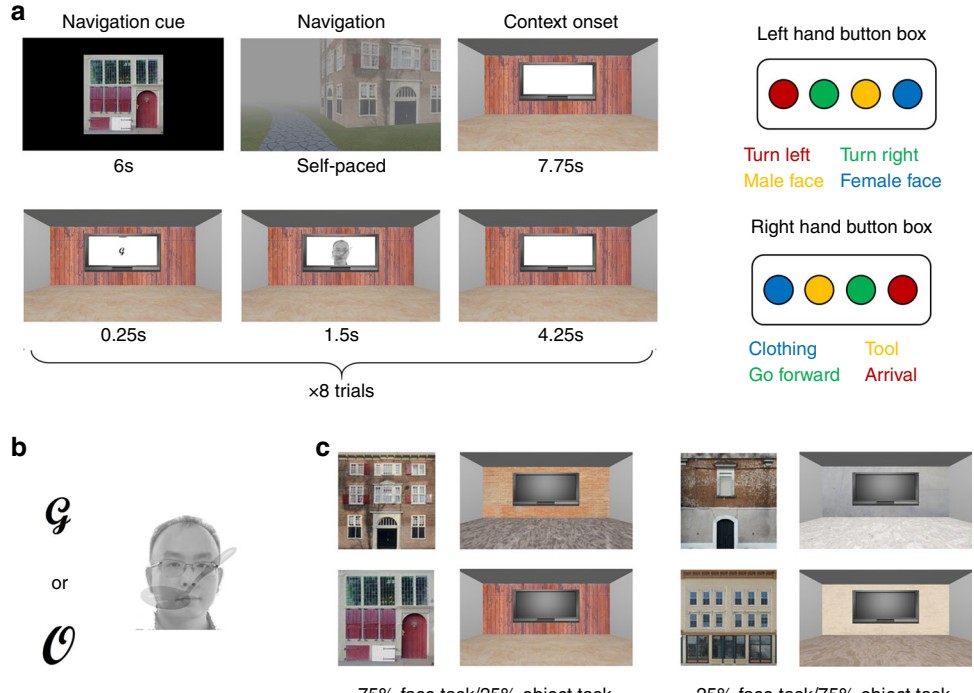

**Fig. 2 Experimental design. a** Experiment trial structure and response mappings. Left: structure of an exemplar block. Participants were first cued to navigate to a target building and then entered a room in the building (top row). They then performed eight perceptual decision-making trials within the room (bottom row). Right: response mapping. Blue, yellow, green and red buttons correspond to index, middle, ring and pinky fingers, respectively. Participants were randomly assigned to use one hand to respond to faces and to use the other hand to respond to objects. **b** Task cues and an example stimulus pair. Depending on the letter/task cue (left), participants were to identify the gender of the face or the type of object in the compound stimulus (right). **c** Building exteriors and room interiors for the four contexts. For each participant, the mapping of the four contexts to the two CTDs was randomly assigned.

$P < 0.00001$, one-sample t-test; $d = 1.58$), consistent with previous findings[23,24]. The prediction error in the temporal model demonstrated a trend towards modulating accuracy (regression coefficient: $0.09 \pm 0.05$; $t_{32} = 1.81$; $P > 0.07$, one-sample t-test; $d = 0.31$). Crucially, analyses further revealed that larger contextual model prediction error predicted lower accuracy (regression coefficient: $-0.19 \pm 0.04$; $t_{32} = -4.61$; $P = 0.00006$, one-sample t-test; $d = 0.80$; Fig. 3b) and slower RT (regression coefficient: $0.01 \pm 0.004$; $t_{32} = 2.62$; $P = 0.01$, one-sample t-test; $d = 0.46$; Fig. 3c, see Supplementary Fig. 1 for individual data). Given that both the contextual model and the temporal model were included in the same analysis to explain variance in the behavioral data, these latter findings indicate that CTD was learned and influenced task execution. Note that FDR correction[25] was applied to control for multiple comparisons in these behavioral analyses, and all reported significant behavioral results survived FDR correction. Repeating the behavioral analyses using residuals, after regressing out shared variance, did not qualitatively change the results (Supplementary Note 1).

**Reinstatement of CTD in frontoparietal cortex.** Having documented the learning of CTD in behavior, fMRI analyses focused on quantifying neural activity patterns related to the representation and retrieval of the CTD. Given prior demonstration that task representations can be decoded from patterns of activity in human frontoparietal cortex[10,11,26–31], BOLD patterns were extracted from frontoparietal regions-of-interest (ROIs) defined using the Human Connectome Project's multi-model cortical parcellation[32]. We hypothesized that predictions (i.e., retrieval) of CTD would occur upon arrival in the context. Accordingly, we computed the pattern similarity between BOLD activity patterns evoked at the onset of

each context (i.e., the 7.75 s pre-task period coinciding with the onset of the room at the beginning of each block; Fig. 2c) and activity patterns evoked at the onset of the task cue on each trial.

Specifically, because learning had yet to occur, data from the first run were excluded from the pattern similarity analyses (see the "Methods" section). For each context in all subsequent runs, the context's pattern similarity was computed in relation to out-of-run trials[33]. Note that the context has an associated task demand defined by its CTD. Moreover, a trial is always presented in a context and hence also inherited the associated task demand from this context (i.e., defined by the CTD of the context). To quantify task representations and their prediction, we computed three types of context-trial pattern similarity: (1) Same Context (i.e., context and trial in the same room that, by definition, have the same associated task demand); (2) Same CTD (i.e., context and trial in different rooms that have the same associated task demand); and (3) Different CTD (i.e., context and trial in different rooms that have different associated task demand). This yielded a 3 (context: Same context, Same CTD, Different CTD) × 2 (congruency: match/mismatch between the CTD and the actual task demands on the trial) factorial design (Fig. 4a, b). We hypothesized that reinstatement of CTD during context onset will bias the neural activity patterns towards the predicted task. Therefore, if the context and the trial are associated with the same task (i.e., same context and same CTD conditions), then context-trial pattern similarity should be higher on match than mismatch trials. On the other hand, if two trials are associated with different tasks, then this effect will be attenuated or reversed, because the unexpected task in one context (e.g., face trials in a 75% object context) is the predicted task in other contexts (e.g., face trials in a 75% face context).

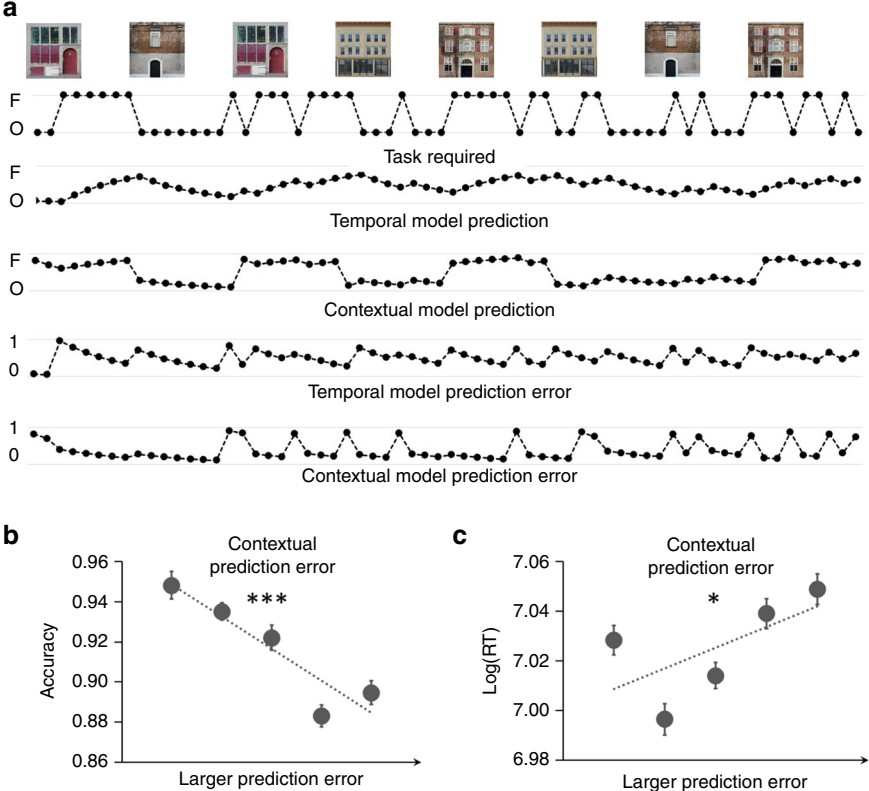

**Fig. 3 Behavioral analyses and results. a** Time courses of reinforcement learning model outputs for an example run. The building images mark the beginning of blocks. Each dot represents one trial. Note that contextual model predictions are more sensitive to change of contexts than temporal model predictions. **b**, **c** Quintiles (*x*-axis) and mean ± SEM of group-level accuracy (**b**) and logged RT (**c**), plotted as a function of contextual model prediction error. *\*p < 0.05, \*\*\*p < 0.001* (two sided one-sample *t*-tests). The experiment was conducted once (*n* = 33 biologically independent samples). *P* values are uncorrected. Source data are provided as a Source Data file.

Given the above logic, we tested whether CTD reinstatement occurred during the onset of a context using a context × congruency interaction (Fig. 4b). Strikingly, four frontoparietal regions––left inferior frontal junction (IFJP), left superior frontal (BA i6-8), right dlPFC/frontopolar (BA 9-46d), and left superior parietal (BA 7PL)––showed a significant context × congruency interaction (*p* < 0.05, FDR corrected; Fig. 4c–f, brain images). In post-hoc analyses, we tested the effect of congruency collapsed across Same Context and Same CTD conditions, as well as in the Different CTD condition. All four ROIs displayed higher context-trial pattern similarity on match than mismatch trials when the associated task demands were the same between the context and the trial, though the effect in left superior parietal did not reach significance (Fig. 4c–f, dot graphs; Table 1; see Supplementary Fig. 2 for individual data). Critically, while it is possible that context-trial pattern similarity on Same Context trials reflects perceptual coding of the room rather than task coding, significant effects in the Same CTD condition must reflect task demands. Importantly, all ROIs displayed a numerical trend of higher context-trial pattern similarity on congruent than incongruent trials in the Same CTD condition, with right dlPFC/frontopolar cortex and left inferior frontal junction reaching statistical significance (Fig. 4c–f, dot graphs; Table 1). Finally, all four ROIs exhibited lower context-trial pattern similarity on match than mismatch trials when the associated task demands differed between the context and the trial, though the effect in right dlPFC/frontopolar cortex did not reach significance (Fig. 4c–f, dot graphs; Table 1).

To further examine whether the cortical reinstatement of CTD predicts behavior, we conducted a trial-level brain-behavior

analysis on each of the four identified frontoparietal ROIs. Specifically, trial-level reinstatement was measured using the match/mismatch contrast on Fig. 4b. The Different CTD condition was excluded from this analysis to avoid the measure of CTD reinstatement being confounded by the mismatch of CTD between the two contexts (e.g., the mismatch of CTD may impact the context-trial pattern similarity and/or its congruency effect). Trial-level reinstatement was then used to predict trial-level RT. Temporal model prediction error, task, and ROI-mean univariate activity were used as regressors of no interest. We hypothesized that reinstatement should facilitate behavioral performance. Out of the four regions (see Supplementary Fig. 2 for individual data for each ROI), significant reinstatement-related modulation on RT was found in right dlPFC/frontopolar (regression coefficient: −0.0062 ± 0.0024; $t_{32}$ = −2.63; *P* = 0.01, one-sample t-test; *d* = 0.46; survived FDR correction across the four ROIs) and left superior parietal (regression coefficient: −0.0045 ± 0.0024; $t_{32}$ = −1.89; *P* = 0.04, one-sample t-test; *d* = 0.33; did not survive FDR correction across the four ROIs). We repeated the analysis above using both contextual prediction error and temporal prediction error as covariates, and observed a marginally significant modulation of dlPFC/frontopolar CTD reinstatement on RT (regression coefficient: −0.0040 ± 0.0022; $t_{32}$ = −1.81; *P* = 0.08, one-sample t-test; *d* = 0.32). This finding provides initial support for the claim that the neural measure of CTD reinstatement explains behavioral data above and beyond what is accounted for in the behavioral models. These results reflect a facilitation effect (Fig. 4c), such that stronger task reinstatement led to faster RT. The left inferior frontal junction (regression coefficient: 0.0013 ± 0.0025; $t_{32}$ = 0.55; *P* > 0.58, one-

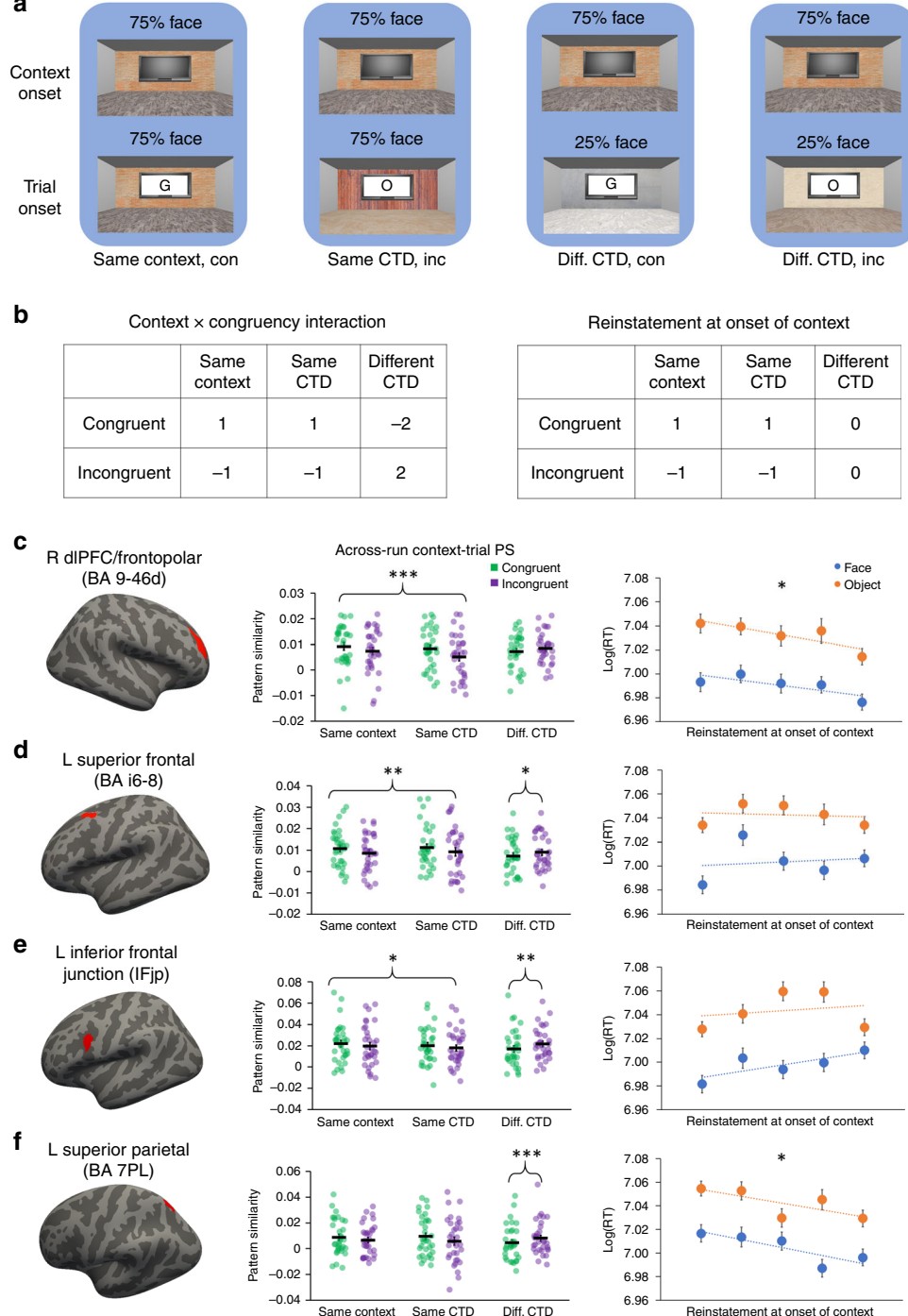

**Fig. 4 Frontoparietal reinstatement of CTD. a** Four examples (grouped by blue background) of how the CTD of context A (top row) and the required task and the CTD of a trial in context B (bottom row) define the experimental condition for testing the reinstatement of CTD (text label below the background). **b** Left: linear contrast testing the reinstatement of CTD. To account for different frequencies of trial types and to make the contrast orthogonal to main effects, the weight for Different CTD conditions was twice the weight for Same Context and Same CTD conditions. Right: contrast used to test the modulation of CTD reinstatement on RT. **c–f** Frontoparietal ROIs showing significant reinstatement of CTD. From left to right: locations of ROI (marked in red), Individual data and mean ± SEM of context-trial pattern similarity plotted as a function of experimental conditions, quintiles (*x*-axis) of group mean of log-transformed RT (±SEM) plotted as a function of task type and context-trial pattern similarity. The names of the ROIs are from the Human Connectome Project's multi-modal cortical parcellation[32]. *$p < 0.05$; **$p < 0.01$; ***$p < 0.001$ (two-sided one-sample *t*-tests). con, congruent; inc, incongruent. For bar graphs, see Table 1 for *P* values. The experiment was conducted once ($n = 33$ biologically independent samples). Source data are provided as a Source Data file.

sample *t*-test) and the left superior frontal (regression coefficient: $-0.0004 \pm 0.0021$; $t_{32} = 0.21$; $P > 0.83$, one-sample *t*-test) ROIs did not reach statistical significance. For completeness, we also repeated this analysis using the congruency effect in the Different

CTD condition; none of the ROIs showed significant modulation on RT in this analysis (all *Ps* > 0.09). We did not see strong evidence for CTD reinstatement in visual, motor, premotor and medial prefrontal cortex (Supplementary Notes 2–4). Together,

**Table 1 Summary statistics of the context-trial pattern similarity congruency effects in the four cortical ROIs showing significant context × congruency interactions in context-trial pattern similarity.**

|  | R BA 9-46d | L BA i6-8 | L IFJP | L BA 7PL |
|---|---|---|---|---|
| Same Context + Same CTD | 0.0026 | 0.0021 | 0.0028 | 0.0023 |
|  | ±0.0007 | ±0.0007 | ±0.0011 | ±0.0012 |
|  | 3.75*** | 3.04** | 2.61* | 1.92 |
|  | $P = 0.0007$ | $P = 0.005$ | $P = 0.01$ | $P = 0.06$ |
| Same CTD | 0.0033 | 0.0020 | 0.0037 | 0.0021 |
|  | ±0.0014 | ±0.0011 | ±0.0014 | ±0.0014 |
|  | 3.65*** | 1.85 | 2.59* | 1.57 |
|  | $P = 0.0009$ | $P = 0.07$ | $P = 0.01$ | $P > 0.12$ |
| Different CTD | -0.0012 | -0.0017 | -0.0036 | -0.0046 |
|  | ±0.0009 | ±0.0008 | ±0.0011 | ±0.0010 |
|  | −1.38 | −2.16* | −3.51** | −4.54*** |
|  | $P > 0.17$ | $P = 0.04$ | $P = 0.001$ | $P = 0.00007$ |

For each condition, the top and bottom rows show the group mean ± SEM and the $t$-value (DOF = 32, two-sided one-sample tests), respectively.
*$p < 0.05$; **$p < 0.01$; ***$p < 0.001$. $P$ values are uncorrected.

these analyses demonstrate that reinstatement of CTD is evident in frontoparietal patterns of activity upon arrival in each context/room, and that the strength of task reinstatement predicts RTs during subsequent task performance.

**Hippocampal pattern similarity predicts CTD reinstatement.** Prior studies document that trial-level estimates of hippocampal univariate activity and hippocampal pattern similarity relate to cortical reinstatement during retrieval[34–38]. To examine the relationship between hippocampal functional activity and (a) frontoparietal indices of CTD reinstatement and (b) the behavioral consequences of CTD, we extracted block-level measures of hippocampal univariate activity at context onset and computed trial-level pattern similarity between hippocampal activity patterns at context onset (i.e., context–context pattern similarity). To account for the possible change in pattern separation after each encounter of a context, context–context pattern similarity was calculated at the onset of the room for each block, again for Same Context, Same CTD, and Different CTD conditions (Fig. 5a). To validate our context–context pattern similarity measures, we found a context-level representation effect, such that context–context pattern similarity within the Same Context condition (0.0063 ± 0.0005) was significantly higher than context–context pattern similarity in the Different CTD condition (0.0044 ± 0.0007, $t_{32} = 2.39$; $P = 0.02$, paired t-test; $d = 0.42$; Supplementary Fig. 3; Note: This test was performed on all 6 runs, because no learning was assumed in this prediction). Context–context pattern similarity did not systematically change as a function of time (Supplementary Note 5).

We next tested for the predicted hippocampal differentiation effect[39–41]–that is, reduced pattern similarity between two distinct events that share a feature (here, the Same CTD condition) compared to pattern similarity between two events with distinct features (here, the Different CTD condition). Consistent with this prediction, hippocampal context–context pattern similarity in the Same CTD condition was significantly lower than that in the Different CTD condition (Same CTD: 0.0031 ± 0.0008; Different CTD: 0.0051 ± 0.0005; $t_{32} = -2.91$; $P = 0.007$, paired t-test; $d = 0.51$; Fig. 5b). Moreover, examination of the temporal profile of hippocampal pattern similarity revealed that similarity was constant across the six scan runs in the Different CTD condition, but significantly declined from Run 1 to Runs 2–6 in the Same CTD (Supplementary Note 6; Supplementary Fig. 4). As such,

hippocampal differentiation for overlapping contexts (i.e., those that shared the same CTD) emerged through time.

Next, we examined whether hippocampal context–context pattern similarity and univariate activity predicted frontoparietal cortical reinstatement of the CTD. Specifically, a block-wise analysis was conducted for each of the four cortical ROIs showing CTD reinstatement effects (Fig. 4c–f). For each ROI, hippocampal context–context pattern similarity in Same Context, Same CTD and Different CTD conditions, along with the hippocampal univariate activity were used as four predictors of block-wise CTD reinstatement in the ROI (calculated using the contrast on Fig. 4b); the univariate activity of the frontoparietal ROI was used as a covariate of no interest. Only the CTD reinstatement in right dlPFC/frontopolar cortex (BA 9-46d) positively co-varied with hippocampal context–context pattern similarity in the Same CTD condition (regression coefficient: 0.083 ± 0.028; $t_{32} = 2.98$; $P = 0.005$, one-sample t-test, $d = 0.52$; survived FDR correction across the three pattern similarity conditions; Fig. 5c, d) and hippocampal univariate activity (regression coefficient: 0.087 ± 0.042; $t_{32} = 2.13$; $P = 0.04$, one-sample t-test, $d = 0.37$; Fig. 5c, d; see Supplementary Fig. 5 for individual data). When repeated separately for the left and right hippocampus, the regression coefficient for the Same CTD context–context pattern similarity did not significantly differ between the two hemispheres (left: 0.037 ± 0.038; right: 0.044 ± 0.032; $t_{32} = 0.12$; $P = 0.9$; paired t-test). Collectively, these results are consistent with theories of pattern completion that posit that the hippocampus drives restatement of associated event features (here the CTD) in cortex (here dlPFC/frontopolar). While fMRI data lack the temporal resolution to definitively demonstrate that the hippocampal effect precedes the CTD reinstatement in PFC, we performed additional control analyses to discount the possibility that the observed hippocampal-PFC relationship was due to a modulation from dlPFC/frontopolar cortex to the hippocampus or to a common modulator (Supplementary Note 7). Again, definitive evidence on this point awaits further experimentation.

We further explored the behavioral relevance of hippocampal activity and representation by examining whether hippocampal univariate activity and hippocampal context–context pattern similarity in Same Context, Same CTD and Different CTD conditions account for trial-level RTs. Because the neural measures were calculated at the retrieval of CTD, a match/mismatch factor was applied to these predictors, similar to the above brain-behavior analysis using frontoparietal ROIs. Hippocampal univariate activity displayed significant modulation on RT (regression coefficient: −0.0044 ± 0.0021; $t_{32} = -2.10$; $P = 0.04$, one-sample t-test; this relationship is marginally significant when including block-wise CTD reinstatement measured in BA 9-46d as a covariate in the model: −0.0043 ± 0.0023; $t_{32} = -1.91$; $P = 0.07$). By contrast, RT was not significantly modulated by context–context pattern similarity in the Same Context (regression coefficient: −0.0005 ± 0.0021 $t_{32} = 0.25$; $P > 0.80$), Same CTD (regression coefficient: −0.0036 ± 0.0023; $t_{32} = -1.56$; $P > 0.12$) and Different CTD conditions (regression coefficient: −0.0012 ± 0.0022; $t_{32} = -0.53$; $P > 0.60$). These results provide suggestive evidence that hippocampal contributions to behavior were partly implemented via hippocampal modulation on CTD reinstatement in right dlPFC/frontopolar cortex.

## Discussion

Retrieval of the task-set that is relevant for impending goals is essential for flexible and efficient behavior. Prior work suggests that retrieval of task demand can be facilitated by the prediction of the relevant task-set in a probabilistic manner[23,42]. In real life, the spatial contexts we live in are often strong predictors of task

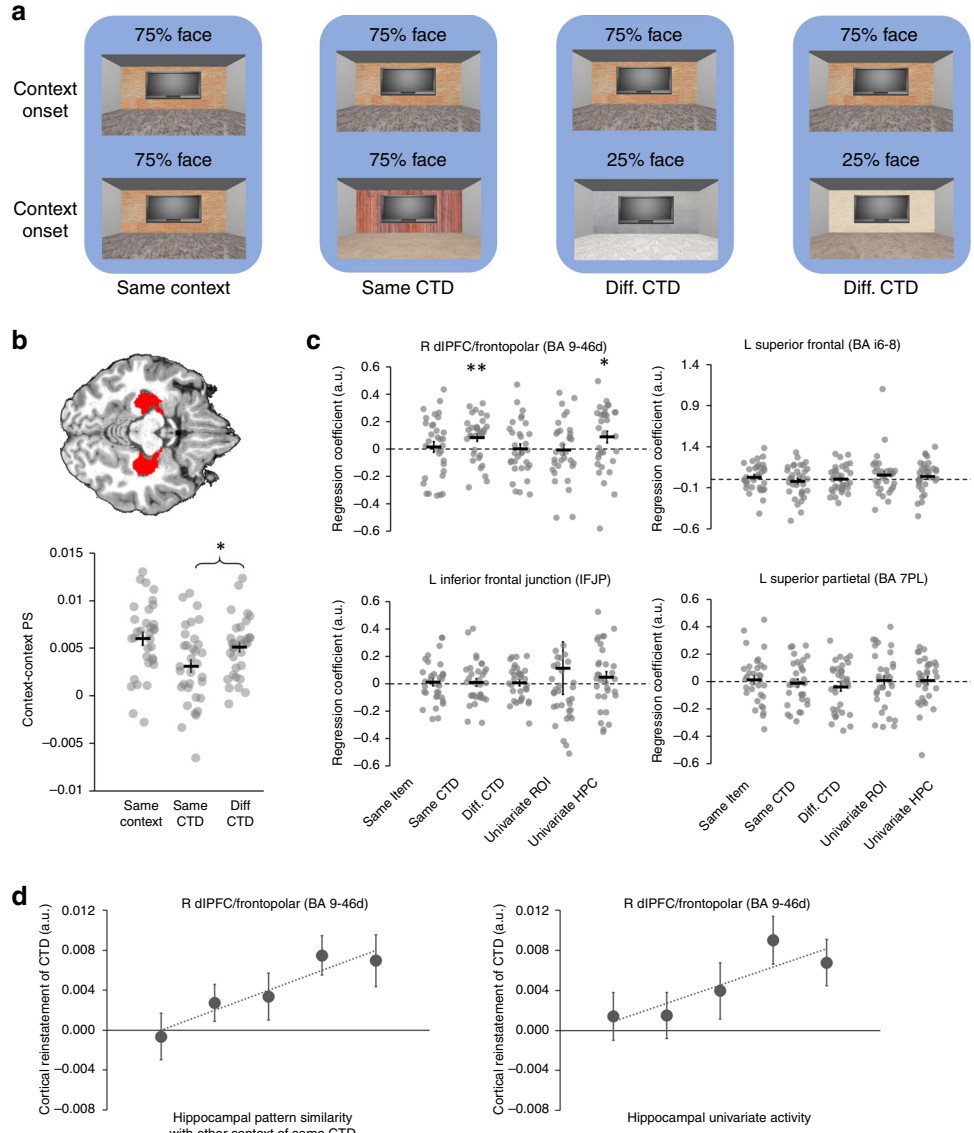

**Fig. 5 Hippocampal activity and pattern separation predict cortical reinstatement of CTD. a** Four examples (grouped by blue background) of how the CTDs of context A (top row) and context B (bottom row) define the experimental condition of context–context pattern similarity (text label below the background). **b** Visualization of the hippocampus ROI (top, in red) and individual data and mean ± SEM of the three conditions of hippocampal context–context pattern similarity (bottom). At the group level, pattern similarity in the Same CTD condition was significantly lower than the Different CTD condition. **c** Individual data and mean ± SEM of the regression coefficients of each of the four predictors and the ROI univariate activity on the CTD reinstatement in each of the four frontoparietal ROIs. **d** Quintiles (x-axis) of group mean of CTD reinstatement in BA 9-46d (±SEM) plotted as a function of hippocampal context–context pattern similarity in the Same CTD condition and hippocampal univariate activity. *p < 0.05; **p < 0.01. P values are uncorrected and from two-sided one-sample t-tests. The experiment was conducted once (n = 33 biologically independent samples). Source data are provided as a Source Data file.

demands. In this study, we investigated the neurocognitive mechanisms that are engaged when spatial contexts cue retrieval of relevant task-sets. To better simulate spatial contexts encountered in real life, participants performed the task in a 3D virtual environment with spatial contexts (i.e., buildings and rooms). These contexts were then associated with different CTDs, which were manipulated by the probabilistic distributions of the task demands required within each context. After controlling for context-insensitive learning of task demand, we observed significant modulation of CTD on both accuracy (Fig. 3b) and RT (Fig. 3c) in behavior, such that behavioral performance was better when the required task matched the predicted task demand.

Given that the spatial context changed after each block, the behavioral results indicate that participants learned and retrieved the probabilistically dominant CTD at each block to guide goal-directed behavior, setting the stage for fMRI analyses focused on the neural mechanisms supporting the retrieval of the associated CTD.

We predicted that neural activity patterns elicited by the reinstatement of the retrieved CTD would be similar to the activity patterns when the task predicted by the CTD was being performed. This prediction was tested using context-trial pattern similarity analyses, which resemble encoding-retrieval similarity analysis in the memory literature[37,43] and complement previous

decoding analyses focused on distinguishing activity patterns elicited by different cued tasks or task demands[23,29,30]. Consistent with previous findings showing instantiation of task-set during task in PFC[5–13], the context-trial pattern similarity analyses revealed frontoparietal foci in which reinstatement of CTD was observed, including right dlPFC (Fig. 4c). Right lateral PFC has been implicated not only in representing task rules, but also in task organizations such as temporal order[27] and composition[26]. We suggest that the reinstatement of CTD in right dlPFC facilitates proactive cognitive control[44], which provides top-down modulation to bias processing towards the task predicted by the CTD. Consistent with this account, the strength of CTD reinstatement in right dlPFC upon entrance into a context covaried with subsequent behavioral performance, such that stronger reinstatement was followed by faster correct responses during task trials encountered in the context (Fig. 4c, quintile graph).

We then investigated the relationship between CTD reinstatement in dlPFC and hippocampal activity at the onset of the spatial context. While CTD reinstatement in dlPFC did not covary with its own univariate activity, it selectively covaried with univariate activity in the hippocampus. This finding is in line with the literature documenting hippocampal-cortical reinstatement coupling during associative retrieval of visual stimuli[34–38]. Multivariate hippocampal activity patterns[45] and connectivity[46] provide additional support for the mechanism of pattern completion during cued retrieval of stimulus-action sequences. The behavioral relevance of the present hippocampal univariate effect is supported by our observation that, as with CTD reinstatement in dlPFC, hippocampal activity at context onset modulates RT on trials subsequently encountered in the block.

To further study the interaction between CTD reinstatement and hippocampal mnemonic mechanisms, we examined the relationship between hippocampal pattern separation and CTD reinstatement in dlPFC. To this end, our analyses were motivated by recent hippocampal findings showing lower pattern similarity between events with shared features than between events with distinct features[39–41]. We first replicated this effect, observing reduced hippocampal pattern similarity between contexts sharing the same CTD as compared to pattern similarity between contexts with different CTDs (Fig. 5b). As with the preceding studies, this finding might to be at odds with pattern separation theory, which posits that pattern separation drives the hippocampal representations of overlapping events to be distinct as would be expected by default for the representations of non-overlapping events. However, such hyper pattern distinctiveness for overlapping events may be explained as a secondary effect that follows initial pattern separation. Specifically, assuming that pattern separation generates initial orthogonal representations for events sharing the same feature, the nonmonotonic plasticity hypothesis proposes that pattern similarity between the events can decrease further through a pruning process that follows weak activation of the overlapping portion of the neural codes[47]. Indeed, analyses of the temporal profile of change in pattern similarity in the Same CTD condition lends support for this latter interpretation, as the hyper pattern distinctiveness (i.e., lower similarity in Same CTD than in Different CTD) emerged over time (Supplementary Note 6). The behavioral relevance of the present hippocampal pattern distinctiveness effect is supported by the observed relationship between hippocampal pattern differentiation in the Same CTD condition and CTD reinstatement in right dlPFC/frontopolar cortex (Fig. 5d), which modulated RT in trials later in the same block (Fig. 4c).

In the present paradigm, we hypothesized that context-level differentiation may support achieving the goal of representing the particular context in which one is situated as well as the goal of remembering the probabilistically dominant task performed

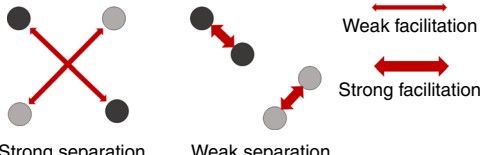

**Fig. 6 Relationship between pattern separation of context representations and CTD retrieval.** Stronger hippocampal separation of contexts sharing the same CTD may lead to more distinct neural coding of contexts (indicated by the distance between dots of the same color), but weaker facilitation from the other context in supporting retrieval of the associated CTD (indicated by the thickness of the red arrows).

within the context. While a priori one might predict that hippocampal pattern separation would similarly benefit both goals, it is possible that the effects can differ. On the one hand, if the goal is to distinguish between two contexts, hippocampal coding of contexts within each pair may be separated to counter the interference between contexts caused by the shared CTD. On the other hand, pattern separation may be disruptive if the goal is to retrieve the associated CTD: Specifically, consider two pairs of contexts (dark and light dots in Fig. 6, each dot representing one context), each associated with one CTD. Relative to weaker hippocampal pattern separation, when pattern separation is strong the cuing of one context is less likely to concurrently retrieve, or suffer interference from, the other context sharing the same CTD. However, while such strong pattern separation might keep the contexts more distinct, this may result in a failure to leverage the other context to boost retrieval and reinstatement of the shared CTD through recurrent retrieval[48,49]. While speculative, the divergent effects on the two goals may explain why overlapping contexts and their shared CTD were, on average, not linked in a single memory trace through integrative encoding[50] (which would have been evidenced by increased pattern similarity between contexts in the Same CTD condition). At the same time, we observed that dlPFC reinstatement of CTD positively scaled with the hippocampal pattern similarity between the two overlapping contexts (i.e., Same CTD condition, see Fig. 5c, d). This observation is broadly consistent with models of hippocampal generalization[48,49], as weaker pattern separation may allow recurrent retrieval of the other context sharing the CTD, which would facilitate the retrieval and reinstatement of the associated CTD. Future research should explore whether and how recurrent or chained retrieval of associated task-sets and sensory information support prospective planning and flexible behavior in complex tasks[51].

The context-task-set model[52] would suggest that participants form a hierarchical structure of the task, with a higher level representing the two CTDs, each of which is further linked to its two associated contexts at a lower level. This theory would appear to predict higher pattern similarity in Same CTD than Different CTD conditions, which is the opposite of the findings in the present study. One possibility is that strong pattern separation at the context level hindered the clustering of contexts sharing the same CTD. More generally, in this experiment, pattern separation at the context level appears to reflect a trade-off between two goals: navigation and context representation that benefited from pattern separation and perceptual decision making that was hindered by pattern separation. Pattern separation may be a dynamic process that optimizes between tradeoffs. Here, these dynamics are consistent with the block-level fluctuations in hippocampal context–context pattern similarity and dlPFC CTD reinstatement. The argument that hippocampal pattern separation balances integration and separation would predict that

stronger pattern separation will lead to stronger facilitation in behavior. We tested this prediction by correlating the Same CTD context–context pattern similarity in the hippocampus with the behavioral modulation of CTD shown in Fig. 3 (separately for accuracy and RT) across participants. Consistent with this prediction, we observed a significant correlation ($r = -0.39$, $P = 0.03$, Supplementary Fig. 6), such that, contextual prediction errors impact accuracy more (indicating stronger behavioral influence of CTD) when the pattern similarity scores are lower (indicating stronger separation).

The present study advances understanding of how the brain retrieves goal-relevant task-sets. Utilizing a 3D virtual environment to induce strong associations between spatial contexts and CTD, we observed that context-CTD associations modulated subsequent goal-directed behavior; these data document a form of task-level memory-guided prospection. The fMRI data further revealed CTD reinstatement in dlPFC (BA 9-46d), which further predicted decision response times at the trial level. This reinstatement of CTD was predicted by hippocampal activity and differences in hippocampal pattern separation between contexts sharing the same CTD. Taken together, these findings document the role of hippocampal-prefrontal interactions in CTD retrieval and goal-directed behavior.

## Methods

**Subjects**. Thirty-eight subjects gave informed written consent, in accordance with procedures approved by the Stanford University Institutional Review board. Two subjects dropped out before the experiment ended (one felt motion sick and one felt anxious). Another two subjects were excluded due to low behavioral performance (accuracy was lower than the group median minus 3 standard deviation). An additional subject was excluded due to excessive head motion. The final sample consisted of 33 participants (18–32 yrs old; 19 females, 13 males and 1 NA) with normal or corrected-to-normal vision and no self-reported history of psychiatric or neurological disorders.

**Behavioral tasks**. Participants performed a perceptual decision-making paradigm (Fig. 2a) embedded in a spatial navigation task in a 3D environment made using Python and Panda EPL. The environment consisted of a circular track and four visually distinct buildings (Fig. 2c) located on the exterior side of the track. The experiment was divided into 6 runs of 8 blocks each, with 2 blocks/building/run. Each block started with a building cue for 6 s, indicating to which building the participant was to navigate. The participant then moved on the circular track to the cued building and indicated their arrival by a button press. If the response was not made at the entrance to the cued building, an error message was presented for 1 s. On error trials, the participant resumed navigating until the correct building was reached.

Immediately upon arriving at the cued building, the interior of a room in the building was presented for 7.75 s (Fig. 2a). The colors and textures of the backwall and the floor were unique for each room, thus creating four distinct spatial contexts––defined by the perceptual features of the room along with which building the room was located. All four contexts included a display on the backwall. For each block within a room, participants performed eight perceptual decision-making trials presented on the display on the backwall. Each trial began with the presentation of a task cue (either the letter G or O) for 250 ms, followed by the presentation of overlapping, translucent images of a face (either male or female) and an object (either clothes or tool) for 1500 ms. The particular combinations of face and object images were randomly generated without repetition for each participant. Depending on the cue, participants were required to categorize either the gender of the face (cued by letter G) or the type of the object (cued by letter O), by pressing one of two response buttons (response mappings are shown in Fig. 2a). Trials were separated by a 4.25 s inter-trial interval. Critically, to induce different learned associations between each context and expected task demands (face vs. object)–i.e., the CTD–blocks consisted of 75% face/25% object trials in two contexts and 75% object/25% face trials in the other two contexts. The pairings of contexts and CTD were randomized across participants. Inspired by the literature of item-cognitive control demand associations[16], we adopted an incidental learning paradigm. However, we do not assume such learning is implicit (or exclusively implicit). We expect that explicitly learned CTD (can result from intentional or incidental learning) will result in similar retrieval processes, given the importance of hippocampus in the retrieval of explicitly formed associations[53]. For example, the hippocampal pattern differentiation findings (c.f., Fig. 5b) were first observed in studies exploring explicit associations[39].

Prior to the fMRI experiment and outside the MR environment, participants were shown all face stimuli (10 males and 10 females) and object stimuli (10 pieces of clothes and 10 tools) for familiarization. Specifically, all stimuli for each category

(e.g., female faces, tools) were displayed on one page in a browser window on the screen. Participants could freely change pages without time constraints. They were instructed that general familiarity would suffice and that there was no need to remember the details of each image. Most participants finished this process within 3 min. Participants then practiced the perceptual decision-making tasks. During this practice, the stimuli (overlapping object and face pairs) were presented on a white background (i.e., no room interior) to prevent the learning of CTDs prior to the fMRI experiment. The practice session consisted of 10 trials for each task, and repeated if the participant failed to reach an overall accuracy of 70%. Participants then practiced the navigation task for 20 trials (5 trials for each building), which allowed them to become familiar with using the button box for navigation as well as to learn the locations of the four buildings within the environment. The structure of a practice navigation trial was identical to the main task (i.e., building cue followed by navigation). However, upon arrival at the correct building, the practice proceeded to the next building cue (rather than moving into a room in the building).

**Behavioral analysis**. We adopted reinforcement learning model-based behavioral analyses[16,23,24]. Without loss of generality, we denote 0 and 1 for a face and an object trial, respectively. The reinforcement learning model learns to predict the task demand at trial $t$ using the probability of performing an object task (denoted as $P_o(t)$, with $P_o(t)$ greater than 0.5 predicting higher likelihood of encountering an object task than face task), in the following manner:

$$P_o(t) = P_o(t-1) + \alpha(T(t-1) - P_o(t-1))$$

where $\alpha$ is the learning rate, and $T(t-1)$ denotes the task performed in the previous trial. Given the trial sequence experienced by a participant, $T$ becomes known to the model. Therefore, given $\alpha$ and a neutral initial prediction $P_o(0)$ of 0.5, we can estimate the prediction of task demand for each trial using the reinforcement learning model. Relatedly, trial-level (unsigned) prediction error, which was used in model-based behavioral analysis, was defined as $|T(t-1) - P_o(t-1)|$, or the absolute difference between $T$ and $P_o$.

We then constructed two reinforcement learning models to separately model (a) temporal predictions and (b) contextual predictions of task demand. The temporal model selectively used temporal information (i.e., ignored context changes), and consisted of only one reinforcement learner that is active throughout the experiment and learns task predictions from the sequence of trials. By contrast, the contextual model used a combination of temporal information and context to learn the CTD, such that there was one reinforcement learner for each of the four contexts. At each trial, only the learner associated with the present context was activated (i.e., no updating of $P_o$ for the other three learners). The four contextual learners shared the same learning rate. Thus, at a given trial, the temporal model and the contextual model differ in: (1) the learning rate used, and (2) the learner updated (for the temporal model it is always the same learner, whereas in the contextual model it is the learner corresponding to the present context). Recent studies have shown the benefit of adopting self-adjusting learning rates in learning models[15,54]. The benefit is more pronounced in changing environments (e.g., when CTD changes over time in the context of the present experimental design). Given that the CTD in the experimental design stayed constant, we chose a simple fixed learning rate. Under this configuration, the learning rate for each model was a free parameter determined by a grid search (range: 0.01–0.99, step size = 0.01): for each combination of learning rates, temporal and contextual prediction errors were calculated using the trial sequence. The prediction errors were regressed against the log-transformed trial-level RTs (see below). Learning rates that led to the lowest errors in the linear regression were used for subsequent behavioral analyses. Note that we did not constrain the signs of the regression coefficients for the prediction errors; therefore, the grid search procedure was neutral to the subsequent analyses. An alternative model using joint temporal and contextual predictions yielded similar behavioral results (Supplementary Note 8).

Normalized prediction errors were used to predict accuracy and RT. A logistical regression was applied to temporal and context prediction errors to predict trial-level accuracy (1 = correct, 0 = incorrect) for each participant. Covariates included the task cue (face or object task) as a categorical covariate and a constant accounting for the grand mean of accuracy. For RT analysis, error trials, post-error trials and RTs longer than 2.5 standard deviations from the median were excluded. Specifically, post-error trials are known to display 'post-error slowing', possibly due to a cautionary shift in response thresholds[55], which represents a process that is not targeted in the model-based behavioral analysis or the retrieval of CTD analysis. For each subject, a linear model including temporal and context prediction errors, task cue and constant was fit to the logarithm of RT to make the distribution of RT more Gaussian[24]. For the accuracy and RT analyses, regression coefficients for both prediction errors were submitted to group-level t-test against 0.

**MR data acquisition**. Data were acquired on a 3T GE Discovery MR750 MRI scanner (GE Healthcare) using a 32-channel radiofrequency receive-only head coil (Nova Medical). Functional data were acquired using a 3-band echo planar imaging (EPI) sequence (acceleration factor = 2) consisting of 63 oblique axial slices parallel to the long axis of the hippocampus (TR = 2 s, TE = 30 ms, flip angle = 74°, FOV = 215 mm × 215 mm, voxel size = 1.8 × 1.8 × 1.8 mm³). To correct for distortions of the B0 field that may occur with EPI, we collected two B0 field maps

before every functional run, one in each phase encoding direction, with the same slice prescription as the functional runs. Structural images were acquired using a T1-weighted (T1w) spoiled gradient recalled echo structural sequence (186 sagittal slices, slice thickness = 0.9 mm, TR = 7.26 ms, FoV = 230 mm × 230 mm, in-plane resolution = 0.9 mm × 0.9 mm).

**Anatomical data preprocessing**. Preprocessing was performed using *fMRIPprep* 1.1.4 (RRID:SCR_016216)[56,57], which is based on *Nipype* 1.1.1 (RRID: SCR_002502)[58,59]. The T1-weighted volume was corrected for intensity non-uniformity (INU) using N4BiasFieldCorrection (ANTs 2.2.0)[60]; a T1w-reference was used throughout the workflow. The T1w-reference was skull-stripped using antsBrainExtraction.sh (ANTs 2.2.0), using OASIS as the target template. Brain surfaces were reconstructed using recon-all (FreeSurfer 6.0.1, RRID:SCR_001847)[61], and the brain mask estimated previously was refined with a custom variation of the method to reconcile ANTs-derived and FreeSurfer-derived segmentations of the cortical gray-matter of Mindboggle (RRID:SCR_002438)[62]. Spatial normalization to the ICBM 152 Nonlinear Asymmetrical template version 2009c (RRID:SCR_008796)[63] was performed through nonlinear registration with antsRegistration (ANTs 2.2.0, RRID:SCR_004757)[64], using brain-extracted versions of both the T1w volume and template. Brain tissue segmentation of cerebrospinal fluid (CSF), white-matter (WM) and gray-matter (GM) was performed on the brain-extracted T1w using FAST (FSL 5.0.9, RRID:SCR_002823)[65].

**Functional data preprocessing**. For each of the 6 BOLD runs per subject, the following preprocessing was performed. First, a reference volume and its skull-stripped version were generated using a custom version of fMRIPprep. A deformation field, to correct for susceptibility distortions, was estimated based on two EPI references with opposing phase-encoding directions, using 3dQwarp (AFNI). Based on the estimated susceptibility distortion, an unwarped BOLD reference was calculated enabling a more accurate co-registration with the anatomical reference. Head-motion parameters with respect to the BOLD reference (transformation matrices, and six corresponding rotation and translation parameters) were estimated before spatiotemporal filtering using MCFLIRT (FSL 5.0.9)[66]. The BOLD time-series were resampled onto their original space by applying a single, composite transform to correct for head-motion and susceptibility distortions. These resampled BOLD time-series will be referred to as preprocessed BOLD in original space, or just preprocessed BOLD. The BOLD reference was then co-registered to the T1w reference using bbregister (FreeSurfer) which implements boundary-based registration[67]. Co-registration was configured with nine degrees of freedom to account for distortions remaining in the BOLD reference. The BOLD time-series were resampled to MNI152NLin2009cAsym standard space, generating a pre-processed BOLD run in MNI152NLin2009cAsym space. Several confounding time-series were calculated based on the preprocessed BOLD: framewise displacement (FD), DVARS and three region-wise global signals. FD and DVARS were calculated for each functional run, both using their implementations in Nipype (following the definitions by Power et al.[68]). The three global signals were extracted within the CSF, WM, and whole-brain masks. The head-motion estimates calculated in the correction step were also placed within the corresponding confounds file. All re-samplings were performed with a single interpolation step by composing all the pertinent transformations (i.e., head-motion transform matrices, susceptibility distortion correction, and co-registrations to anatomical and template spaces). Gridded (volumetric) resamplings were performed using antsApplyTransforms (ANTs), configured with Lanczos interpolation to minimize the smoothing effects of other kernels[69]. The preprocessed fMRI data were smoothed by a 2 mm full-width-half-maximum Gaussian kernel.

**fMRI analysis**. Prior to fMRI analyses, we removed the first 5 TRs in each run. For each run, we then built a general linear model, which was then regressed against preprocessed fMRI data at the voxel level. To obtain event-level beta estimates for brain activity, each event (i.e., onset of building cue/building exterior and onset of room interior for each block, and onset of task cue for each trial) was represented by a single regressor of a hemodynamic function time-locked to the onset of the event. Each event was modeled using a stick function, because it was a priori unclear whether the learning and retrieval of CTD would last for the whole duration of stimulus presentation. The GLM also included nuisance regressors marking outlier TRs (DVARS > 5 or FD > 0.9 mm from previous TR) and regressors representing TR-level 6-dimensional head movement estimates and global mean signals within the whole brain, WM and CSF masks. The GLM was then regressed against the fMRI data at each voxel, yielding estimates of event-level brain activity. Trials excluded from behavioral analyses and trials with onset time within 12 s prior to an outlier TR were excluded from fMRI analyses (15 ± 6 trials per subject). Due to the focus on retrieval in this study, we excluded data from the first run based on the approximation that a low learning rate of 0.02 will lead to a clear difference of ~0.32 between the two CTDs after one run. A single GLM was constructed for each of the five runs.

For each participant, we conducted pattern similarity analyses on data extracted from bilateral hippocampus (defined by FreeSurfer) and 150 lateralized frontoparietal and medial temporal ROIs (defined by the multi-modal cortical parcellation from the Human Connectome Project; major assignment IDs: 13, 16,

17, 19, 20, 21, and 22, ROI size = 198 ± 16 voxels; range = 51–511)[32]. For each ROI and each event, its activity pattern was quantified as a vector of multi-voxel normalized betas by dividing the original betas by the square root of the covariance matrix of the error terms from the GLM estimation[70]. All voxels in the ROIs were used in the calculation of pattern similarity. Pattern similarity was measured as Fisher transformed Pearson's r. All pattern similarity analyses were conducted across runs, in order to avoid artifacts[33]. Multiple comparisons were controlled for using FDR correction[25]. Unless otherwise specified, all reported P-values were uncorrected. We also report whether the reported results survived FDR correction if applicable. All pattern similarity analyses and statistical tests were conducted using Matlab 2017a.

**Reporting summary**. Further information on research design is available in the Nature Research Reporting Summary linked to this article.

## Data availability
Raw MRI data and behavioral data can be downloaded at [https://openneuro.org/datasets/ds002169]. A reporting summary for this Article is available as a Supplementary Information file. The source data underlying Figs. 3b, c, 4c–f, 5c, d and Supplementary Figs. 1–6 are provided as a source data file.

## Code availability
Analysis scripts are available at [https://github.com/JiefengJiang/CTD].

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

## Acknowledgements

This project was supported by the National Institute on Aging (F32AG056080 to J.J.; R21AG058111 to A.D.W). We thank Kevin P. Madore for helpful comments on a previous version of this paper.

## Author contributions

Conceptualization: J.J. and A.D.W.; Methodology: J.J., S-F.W., W.G., and C.F.; Software: J.J. and C.F.; Formal analysis: J.J., S-F.W., and W.G.; Data curation: J.J.; Writing original draft: J.J.; Writing, review, and editing: S-F.W., C.F., and A.D.W.; Funding: J.J. and A.D.W.; Supervision: A.D.W.

## Competing interests

The authors declare no competing interests.
