## [Peer Review File · Nature Communications]

Reviewers' comments:

Reviewer #1 (Remarks to the Author):

In this study, authors demonstrated the interactive process for retrieving the associated task in a given context (contextual task demand, CTD) via implicit probability learning. They aimed to link three different processes with a hypothesized model which posits: (1) a spatial context will first trigger hippocampus to retrieve its associated CTD (i.e., via context-CTD association), (2) the CTD will be reinstated in the PFC, and (3) this process will predict goal-directed behavior (i.e., accuracy and RT on a perceptual discrimination task). To demonstrate this framework, the authors showed the results that (1) more prediction error (PE) was linked to slower RT and lower accuracy for the cue-based task, showing CTD learning; (2) context-trial pattern similarity (PS) for the congruent (i.e., same CTD in context and trial periods) was higher than for the incongruent condition in prefrontal sub-regions, which was interpreted as reinstatement of CTD, which was also negatively related to RT; (3) and context-context PS for the same CTD in different contexts and univariate activation in hippocampus were positively related to CTD reinstatement in PFC.

The research question in this study is important, and the study design is clever and well-suited for the goal. However, I have concerns about the data analyses and their interpretation which weaken the overall arguments of the paper. Further, I found the writing style to be arduous on the reader – too many acronyms (e.g., CTD, PS, PE, ...) used too frequently – that made it challenging to parse the results and discussion alike. More handholding of the reader would benefit the impact of this paper. Note: I will use the authors acronyms in my comments for consistency/clarity.

Major points

1. The authors emphasize the importance of implicit learning of contextual task demand (CTD) over time via mnemonic mechanisms (line 47). However, it was not clearly shown that the statistical learning was built up as a function of time (or repetition). It would be helpful if there was a clearer foundation as to why you focused on implicit learning compared to explicit learning when people learned the CTDs. The benefit here would be that implicit learning might, I assume, create more continuous scales (or variance) for the RT given that CTD would have been gradually learned. If so, the learning effect should be provided for the (a) behavioral outcomes, (b) pattern separation in hippocampus for same CTD, and the (c) relationship between dlPFC and (b) or (a) as a function of repetition. Or at least, the differences between the first run, which was removed from the main analysis, and the subsequent runs should be provided.
2. Do you believe that the CTD retrieval process would be different if the association was explicitly learned compared to it was learned implicitly? Although implicit learning was introduced as an important topic, I couldn't find the implication for that topic in the results.
3. For the pattern separation account for CTD learning, the authors set up the argument (line 82) that

CTD learning would promote pattern separation between the different contexts that share same CTD to prevent retroactive interference from recurrent retrieval (i.e., co-activated context). This argument should be verified if the pattern separation increased in the course of the learning. Or, even if the learning was fluctuated rather than linearly improved over time, the relationship between CTD learning effect (e.g., RT or accuracy) and the pattern separation should be provided to support the argument.

Moreover, it is strange to use the difference between the Same CTD with different contexts vs. Different CTD as a validation for context-context PS measurement (in Figure 5 results). To verify if the context-context PS in hippocampus could successfully measure the representation of the context-CTD association, the PS in Same context with same CTD should be significantly higher than the PS between Different CTDs in different contexts. Indeed, it was not clearly verified that Same CTD with different contexts increased pattern separation over the learning.

4. (Figure 3 results) Although the contextual model seems to predict CTD better than temporal model (which needs quantitative comparisons), the behavioral results (line 150) show that the regression coefficient for the temporal model PE (0.32 ± 0.004) was higher than that for the contextual model PE (0.01 ± 0.004) for RT, suggesting that temporal model had a better predictability for the RT (i.e., the bigger temporal PE, the slower RT). Given that the contextual model includes both temporal and contextual information, how could you conclude that the CTD was learned through contextual information alone? Moreover, it should be explained clearer (or more explicitly) why the model prediction (or PE) would be more sensitive to detect CTD learning effect compared to behavior outcome itself.

5. The formula " $Po(t) = Po(t) + \alpha(T(t-1) - Po(t-1))$ " is not mathematically valid, as one of $Po(t)$ in both hands should be replaced based on the model setup. For example, if the prediction is integrated, then the denotation should be $Po(t-1)$ on the right hand of the formula. More details for each element will be also helpful. For example, if the Po is over 0.5, does it predict the object task more, and vice versa for the face task (i.e., under 0.5)? Does the $T(t-1) - Po(t-1)$ indicate PE? Then, shouldn't it be $|T(t-1) - Po(t-1)|$ in the formula? Or PE was separately calculated? Also, it needs more details as to how the reinforcement learning model was differently built for the temporal model (using only trial sequence) vs. contextual model (using contextual information + trial sequence, in Methods, behavioral analysis, line 470). Was the only difference which trials were fed into the model for learning?

6. Fig. 5D shows that context-context PS in hippocampus for the same CTD in different contexts predicts better CTD reinstatement in the prefrontal area, which means that less pattern separation for the different contexts sharing same CTD predicts better CTD reinstatement. Isn't it the opposite from the prediction in the introduction (line 84)? The authors covered this opposite result in the discussion; however, it was still not justified enough how the inconsistent results could support the same argument.

7. In the model in the Fig. 1A, the process has one directional modulation from the hippocampus to behavioral outcome. I am curious if you can show the directional relationship for the CTD retrieval?

Minor points

1. In Fig.1B, it would be helpful if the figure shows how the separated representation predicts CTD learning.
2. (line 150) Was there no statistical significance for the accuracy in the temporal model?
3. The scale for the x-axis (PE) in the Figure 3 was missing. Did you group the PE in 5 scales or were there only 5 levels of PE? Why was the RT transformed to logged scale? This was never explained or justified. Also, the relationship in the Fig 3C seems to fit better for U-shape.
4. What does “the context’s PS was computed in relation to out-of-run trials (line 179)” mean? Did you mean that you did not include PS within a run to remove auto-correlation?
5. It would be easier to read the plots with significance symbols (asterisks) in Fig. 4C-F, middle column.
6. In the Fig. 4B, right panel, was the data being used for the contrast still context-trial PS? Also, were the same task/other task same trials with congruent/incongruent on the left panel? And what is the scale of reinstatement at onset of context on the x-axis?
7. (line 183) Does “context and trial ...” means RSA between context and trial timepoints, same as “context-context RSA” for RSA in context timepoints between paired contexts? Then, using context-trial RSA would be more consistent.
8. (line 274) What does “block-level” mean? Does it mean that the pattern was averaged within a block or the PS was conducted across blocks rather than within a block?
9. In Fig. 5C-D, it seems the term “modulation index (a.u.)” was not defined. Also, the scale on the x-axis in panel D were missing.
10. (line 500) Why were the post-error trials excluded?
11. (line 564) Was the separate GLM built for each trial or were all regressors with each regressor for each trial built in one GLM?
12. For pattern similarity analysis, what did you mean by normalized beta? Does it mean voxel-wise normalization of the beta estimate modeled on each trial? Also, did you use all voxels under each defined ROIs? Then, what was the number of voxels selected for each PS?
13. Was the context also modeled separately for the beta patterns? And given that it was presented for 7.5 seconds, was it modeled with boxcars?

14. For the Methods, while the preprocessing for the imaging data was described so detailed (maybe too specific), other analyses were not described enough.

Reviewer #2 (Remarks to the Author):

Summary:

In the present study, Jiang and colleagues use reinforcement learning models and fMRI to investigate performance in a cue-guided, context-dependent perceptual decision task. Participants view a “navigation cue” that guides them to one of four virtual “rooms”. Perceptual trials within these rooms consist of stimuli that are more or less difficult for each task. They then are shown a “task cue” which indicates which type of judgement they will try to make about the upcoming perceptual stream - gender or object type. Then, they perform eight trials of the perceptual task.

The authors report that a context-aware RL model explains additional variability in behavior unexplained by a context-unaware model. They also report that patterns of fMRI activity in hippocampus and DLPFC at the time of cue presentation predict performance in the subsequent perceptual decision task. Specifically, they report that the differentiation of hippocampal activity patterns between different cues that point to the same room and perceptual task predicts the fidelity of reinstatement of the “contextual task demand” in DLPFC, which subsequently predicts behavioral performance.

While the experiment and analyses reported are quite impressive from a methodological standpoint, it is unclear what precisely is learned from the results. Importantly, there are unresolved questions about what appears to be the primary claim of the paper, the directional relationship between hippocampus, DLPFC, and behavior (illustrated in Figure 1a).

Major:

1. We found it difficult to distill the primary claim of the paper. It appears to be that hippocampal activity, indicated by pattern separability, indexes reinstatement of “contextual task demands” in DLPFC, which subsequently modulate behavior. The directionality and causal nature of the relationship appear to be central to the claim - the manuscript is peppered with language such as that “hippocampal representations of context modulate proactive retrieval” (Line 38) and “CTD depends on the hippocampus” (Line 73).

If this is the main finding, it leads to two concerns:

- First, it’s not clear what this finding would add to previous work, including work from the author’s own lab. Specifically, it has been shown that contextual task demands (stimulus-action tendencies) are reinstated by hippocampus on the basis of informative cues (Hindy & Turk-Browne 2016, 2019), and that

contextual task demands can be decoded from PFC activity patterns preceding performance of the task, and used to predict performance on the subsequent task (Waskom et al 2014, 2016, 2017). Moreover, prior work has shown task-dependent tuning of sensory processing (Tajima et al., 2017) and sensorimotor processing can be tuned according to the anticipated demands (Muhle-Karbe et al., 2017). Given these findings, it would be helpful if the authors could pull reinstatement results from visual ROIs (e.g., face area and object area and/or V1) as control regions compared to the DLPFC to rule out perceptual facilitation and 2) include a discussion explicating how their neural findings are similar and/or distinct from, e.g., the Hindy findings.

- Second, and more importantly, it is not clear that this chain of events is supported by the analyses presented. Specifically, the claim would require:

a. That hippocampal reinstatement precede DLPFC reinstatement. However, both activity patterns are decoded during the same time interval. No evidence in support of the necessary ordering is presented - and this would be difficult to do in fMRI. In fact, there is evidence in the literature of the opposite order of cue-evoked processing, though not in the exact same task (Hung et al 2013).

b. Excluding the possibility that correlations between hippocampal activity, DLPFC activity, and behavior are driven by a common, unobserved, cause. Here, evidence for this is that the behavioral model predictions are relatively improved (but see below for technical questions regarding this analysis) by the addition of HC and DLPFC reactivation measures. This does not exclude the possibility that all three are reflective of variability in learning performance that is not captured by the model, but which originates from neural processing unrelated to these regions. The relationship between activity and behavior is itself unconvincing, because functions commonly ascribed to these regions are also reflective of underlying causes that could occur in parallel to the behavioral variability of interest (e.g. stimulus surprise or novelty detection, pre-fetching of upcoming stimuli or stimulus-action associations, working memory contents or general load).

2. A very intriguing claim, somewhat obscured in the manuscript (Discussion, Lines 398-402) is that pattern separation entails a tradeoff against the cost of reduced transfer learning between instances of the same task that were preceded by different cues. This would be an interesting result, and should be highlighted earlier on in the paper. However, it needs further substantiation, linking pattern similarity measures to subsequent performance improvements (or decrements). For instance: is it the case that pattern similarity following cue A at timepoint t predicts improved performance (on match trials) on that same task at later timepoints $t+x$ (and worsened performance on mismatch trials), following Same-CTD cue B?

3. There are several places in the manuscript where the authors use multiple regression to support the claim that covariate B accounts for variance unexplained by covariate A - e.g. in the comparison of the temporal and contextual models, and in the However, this approach is only valid when the covariates are perfectly orthogonal. Can the authors report the correlation between regressors, and whether the results hold if the regressors are orthogonalized against each other, and/or the regression of each is run on the residuals after regression on the other?

- In the case of the model comparison, an alternative approach could be to combine the two models into

a single model, for instance by using a linear weighting parameter (e.g. $w \cdot \text{temporal} + (1-w) \cdot \text{contextual}$). This would allow them to compete for prediction of behavior on equal terms and permit statistical analysis commensurate with the regression weights reported, while still providing separable estimates of trial-by-trial activity.

- It seems as though a strong test of the author's hypothesis might be that the pattern similarity measure predicts the value of this weighting parameter, across trials or runs. There are many reasons that this wouldn't work, but it may be worth investigating.

- In general, the authors claims would be better supported if they could show that some neurally-derived measure improves the predictive power of the behavioral model.

- Is it the case that the contextual model fits behavior "better" than the temporal model? Or vice-versa? If the claim is that the contextual model predictions underlie the neural activity, and on this basis the temporal model is excluded from MRI analysis, it seems important to justify this exclusion - either by showing a better fit to behavior, or by identifying separable neural correlates.

- Can the authors present a comparison of the contextual (and temporal) models against a model that allows learning rate to vary between contexts? This might have been a minor comment, but for the above query about whether neural responding and behavior may be both driven by learning variability not captured by the model.

Minor:

1. We suggest that the authors reconsider using "PS" to refer to pattern similarity, as the manuscript also relies on the term "pattern separation" or "pattern separability" at several points. This confused us in more than one instance. Perhaps "PSim" might be a better abbreviation?

2. The labels of the regions in each MRI image are a bit obscure, especially in contrast to the more simplified names used elsewhere in the manuscript. Can the authors please choose descriptive labels and use them consistently throughout the manuscript?

3. The use of a grid-fitting approach is strange. While we don't necessarily suspect that this introduced any confounds, why don't the authors employ a more standard optimization approach?

4. The operational definition of pattern similarity is intriguing and may obscure some issues. Specifically, similarity is defined across runs (Page 9, Lines 178-179), which might lead to a confound with learning, as similarity should only decrease across time. Is there a common reference point against which pattern similarity can be measured? Or could the authors instead employ a dissimilarity measure (e.g. against patterns evoked by the other cues within the same run)?

5. Page 20, Line 474: The left side of the equation should read $P_o(t+1)$.

6. Page 24, Line 578: Activity was extracted from bilateral hippocampus, were the final effects with DLPFC driven by the right hippocampus?

7. Is the “contextual task demand” distinct from a response bias? That is, the authors show that RT and accuracy are correlated with the “prediction error” derived from their models, and suggest that this is indicative of ongoing conflict between task performance and reinstated task demands. However, this prediction error is exactly one minus the predicted probability of the task, which has associated with it a pair of responses distinct from those of the other task. Does this effect persist throughout all eight trials in each room? Or is it largely found in the first few, and then decays, as might be expected of a response bias?

Reviewer #3 (Remarks to the Author):

Jiang and colleagues present a carefully designed study that investigates the interactions of prefrontal reinstatement of task demands and hippocampal activity during a novel immersive virtual navigation paradigm. Using a reinforcement learning model they show that a model that learns the task-demands of the contexts tracks subject behavior well. Pattern similarity analyses during the task revealed that participants who reinstate representations of the context’s task demands, respond more quickly in a perceptual decision-making task. This effect was strongest in a dorsolateral prefrontal cortex (dlPFC) ROI and a superior parietal ROI. The authors document a pattern separation effect in HPC such that contexts that have different task demands show higher similarity than contexts that share a task demand. Finally, they examine the relationship between prefrontal task pattern reinstatements and hippocampal activations and pattern similarity. They show a positive relationship between dlPFC and hippocampal task reinstatement and univariate activity that is specific to when two contexts shared the same task demands.

The study is well designed and executed. This paper makes important advances in understanding how the hippocampus and regions in the prefrontal cortex support contextual retrieval of relevant task demands. Though I have no serious problems with the manuscript, I have a few concerns outlined below regarding additional analyses, clarifying certain experimental choices made, and improving the transparency and reproducibility of the work presented.

1. Individual subject data - Throughout the paper the authors report quintiles for their effects. This approach nicely illustrates their main findings, however, it would be useful to show the individual subject data for these analyses. This is a within subjects design and showing the individual subject data is important for understanding variability in the effect. Furthermore, the authors state the first run of scanning was thrown out because subjects had not yet learned the task. It would be nice to include

learning curves for individual subjects to illustrate this effect.

2. Medial ROIs - mPFC has been shown to be important for context dependent memory retrieval and certain aspects of generalization behavior. It could be of interest for the authors to examine pattern similarity in this regions for the Same CTD condition and how that relates the dlPFC effects reported in the manuscript.

3. It may be of interest to the authors to examine the relationship between univariate activity in motor regions and the dlPFC same CTD pattern similarity effect. Given that the task sets are paired with different handed button responses, we might expect an effect where when subjects enter a context you get modulation of activity in motor regions in a task dependent way. This analysis could expand on and further support the hypothesis that dlPFC is contributing to retrieval of task-sets via a suppressive mechanism.

Minor suggestions

- You have chosen to use the human connectome project parcellation for your regions of interest in this paper. Some of these ROIs are quite small. It would be useful to report the number of voxels in each ROI on average.
- Regarding the above point, in the manuscript the authors mention 4 ROIs that were used, but the reports lateralized effects. Please be clear that lateralized ROIs were used in the main text.
- Table 1 shows four prefrontal ROIs but statistics reported in the main text report lateralized effects. Please make sure these statistics are aligned.
- Correction for multiple comparisons are applied in some parts of the manuscript but not all. Please be explicit about where they are applied by including corrected or uncorrected next to the reported p-value.
- It is important for reproducibility that the authors report the software and version used for statistical analyses performed on the pattern similarity data.
- Presumably the star and the bracket in Figure 5b reflects a significant between-condition difference, but I didn't see this stated in the caption
- I am not sure that "pattern separation" is the best way to describe the hippocampal results. I realize that this is true in the literal sense (i.e., the hypotheses concern differences in fMRI pattern similarity), the authors are referring to theories suggesting that the hippocampus assigns sparse, minimally overlapping representations to similar inputs. While it is true that the hippocampus differentiates between the Same context trial pairs and other conditions, the exaggerated reduction of pattern similarity for the Same CTD trials does not necessarily fit with traditional theories of pattern separation (e.g., Guzowski et al., 2004). By describing this effect in terms of pattern separation, I think the authors are making the data sound more prosaic than they really are—the finding of exaggerated differentiation for similar contexts/items is well-documented (e.g., work by Brice Kuhl's lab) and it is a finding that isn't easily explained by sparse coding alone. There are theories that try to account for this kind of effect (e.g., Ritvo et al., 2019) that may be worthy of consideration.
- Related to above, the authors cite McClelland, McNaughton, and O'Reilly in reference to pattern separation. I may be misremembering, but I did not think that paper emphasized pattern separation and pattern completion, I thought this was emphasized in O'Reilly's later, more biologically-based models (as well as work by Marr, Ed Rolls, etc.).
- I also felt that this statement was overly speculative given what the authors actually found in the

study: “The reduction in hippocampal PS in the Same CTD condition may facilitate the separation of and memory for individual contexts, as suggested by prior work documenting that greater hippocampal pattern distinctiveness is associated with better subsequent memory performance at the item level”

Reviewed by Charan Ranganath in collaboration with a trainee
(I sign all reviews)

Response to reviews

We thank Dr. Ranganath and the two anonymous reviewers for their positive feedback and thoughtful comments, based on which we extensively revised the manuscript. We deeply appreciate the input, as we believe the work was strengthened through revision. Please see below our point-by-point responses to the comments.

Reviewer #1

Further, I found the writing style to be arduous on the reader – too many acronyms (e.g., CTD, PS, PE, ...) used too frequently – that made it challenging to parse the results and discussion alike. More handholding of the reader would benefit the impact of this paper.

Response: We replaced the acronyms “PS” and “PE” with “pattern similarity” and “prediction error” in the revised manuscript. “CTD” is kept as the only specialized acronym (as compared to other commonly used acronyms such as “RT” and “PFC”).

1) *The authors emphasize the importance of implicit learning of contextual task demand (CTD) over time via mnemonic mechanisms (line 47). However, it was not clearly shown that the statistical learning was built up as a function of time (or repetition). It would be helpful if there was a clearer foundation as to why you focused on implicit learning compared to explicit learning when people learned the CTDs. The benefit here would be that implicit learning might, I assume, create more continuous scales (or variance) for the RT given that CTD would have been gradually learned. If so, the learning effect should be provided for the (a) behavioral outcomes, (b) pattern separation in hippocampus for same CTD, and the (c) relationship between dIPFC and (b) or (a) as a function of repetition. Or at least, the differences between the first run, which was removed from the main analysis, and the subsequent runs should be provided.*

Response: We thank the reviewer for highlighting the potentially interesting question of whether the CTD association strengths might vary across implicit vs explicit learning conditions. In the manuscript, we do not state a perspective on this and regret that the reviewer concluded that implicit learning is central to our mechanistic claims. We could indeed imagine that implicit learning approach would give rise to a continuous strength distribution that shift with learning, we also note that some explicit learning/declarative memory models would make the same prediction. To avoid this potential confusion, we now state that we adopted an incidental learning paradigm as this is typical in the literature on item-specific cognitive control demand learning, but that we do not assume such learning is implicit (or exclusively implicit). The related text has been added to the revised manuscript:

“We adopted an incidental learning paradigm, as this is typical in the literature of item-specific learning of cognitive control demand. However, we do not assume such learning is implicit (or exclusively implicit).” (Page 22)

Following the reviewer’s guidance, we now report the differences between the first run and subsequent runs (for details, see also our response below to point 3a).

2) *Do you believe that the CTD retrieval process would be different if the association was explicitly learned compared to it was learned implicitly? Although implicit learning was introduced as an important topic, I couldn't find the implication for that topic in the results.*

Response: We added the following discussion to the revised manuscript:

“This study used associations that are learned implicitly. We expect that explicitly learned CTD will result in similar retrieval processes, given the importance of hippocampus in the retrieval of explicitly formed associations¹. For example, the hippocampal pattern differentiation findings (c.f., Fig. 5B) were first observed in studies exploring explicit associations².” (Page 21).

3a) *For the pattern separation account for CTD learning, the authors set up the argument (line 82) that CTD learning would promote pattern separation between the different contexts that share same CTD to prevent retroactive interference from recurrent retrieval (i.e., co-activated context). This argument should be verified if the pattern separation increased in the course of the learning. Or, even if the learning was fluctuated rather than linearly improved over time, the relationship between CTD learning effect (e.g., RT or accuracy) and the pattern separation should be provided to support the argument.*

Response: Following the reviewer's guidance, we performed the following test for the temporal change of hippocampal pattern differentiation:

“A key assumption of hippocampal pattern differentiation is that the differentiation occurs, potentially relatively rapidly, through learning. To test this, we calculated context-context pattern similarity between run 1 and runs 2-6 and compared it to the context-context pattern similarity calculated within runs 2-6. Consistent with the pattern differentiation analysis in Fig. 5B, the comparison was performed between the Same CTD and the Different CTD conditions. To test whether the difference in context-context pattern similarity evolves through time, we conducted a repeated-measures 2 (condition: Same CTD/Different CTD) × 2 (time: run 1/run 2-6) ANOVA, which revealed a significant interaction between condition and time ($F_{1,32}=11.22$, $P = 0.002$). Post-hoc analysis showed that in run 1, context-context pattern similarity was marginally higher in the Same CTD (0.0061 ± 0.0010) than the Different CTD condition (0.0043 ± 0.0008 , $t_{32} = 1.84$; $P = 0.07$; paired t-test; $d = 0.33$; Supplementary Fig. 4), whereas in runs 2-6, the pattern was reversed and showed differentiation in the Same CTD condition (Same CTD: 0.0031 ± 0.0008 ; Different CTD: 0.0051 ± 0.0005 ; $t_{32} = -2.91$; $P < 0.01$, paired t-test; $d = 0.51$; Fig. 5B and Supplementary Fig. 4). Taken together, these results support the conclusion that hippocampal pattern differentiation increased through time.” (Supplementary Note 6)

Supplementary Figure 4. Individual context-context pattern similarity, plotted as a function of experimental condition (Same Context and Different CTD) and time (run 1 and runs 2-6). Each line represents one participant.

3b) *Moreover, it is strange to use the difference between the Same CTD with different contexts vs. Different CTD as a validation for context-context PS measurement (in Figure 5 results). To verify if the context-context PS in hippocampus could successfully measure the representation of the context-CTD association, the PS in Same context with same CTD should be significantly higher than the PS between Different CTDs in different contexts. Indeed, it was not clearly verified that Same CTD with different contexts increased pattern separation over the learning.*

Response: The validation results reported in Fig. 5B are motivated by the hippocampal pattern differentiation approach and outcomes in Favila et al², such that pattern similarity between different scenes decreased when the scenes were paired with the same face image than different face images. Because context-context pattern similarity in Different CTD conditions were always calculated between different contexts, we matched this in the Same CTD condition by only using context-context pattern similarity between different contexts.

We thank the reviewer for suggesting another test to validate the hippocampal pattern similarity data. We conducted the test and added the following to the revised manuscript:

“... we found a context-level representation effect, such that context-context pattern similarity within the same context condition (0.0063 ± 0.0005) was significantly higher than context-context pattern similarity in the Different CTD condition (0.0044 ± 0.0007 , $t_{32} = 2.39$; $P = 0.02$, paired t-test; $d = 0.42$; Supplementary Fig. 3; Note: This test was performed on all 6 runs, because no learning was assumed in this prediction).” (Page 13, prior to Fig. 5B results)

Supplementary Figure 3. Individual context-context pattern similarity for Same Context and Different CTD conditions. Each line represents one participant.

4) *(Figure 3 results) Although the contextual model seems to predict CTD better than temporal model (which needs quantitative comparisons), the behavioral results (line 150) show that the regression coefficient for the temporal model PE (0.32 ± 0.004) was higher than that for the contextual model PE (0.01 ± 0.004) for RT, suggesting that temporal model had a better predictability for the RT (i.e., the bigger temporal PE, the slower RT). Given that the contextual model includes both temporal and contextual information, how could you conclude that the CTD was learned through contextual information alone? Moreover, it should be explained clearer (or more explicitly) why the model prediction (or PE) would be more sensitive to detect CTD learning effect compared to behavior outcome itself.*

Response: Thank you for this important suggestion. We now clarified the rationale behind the model-based behavioral analysis in the revised manuscript, which reads:

“A seemingly straightforward way to test the learning of CTD would be to compare the behavioral performance between conditions when the required task was congruent with the CTD (e.g., face task in a context of 75% face trials) to when it was incongruent (e.g., face task in a context of 75% object trials). However, this test can be confounded by other learning strategies that (partially) capture the statistical contingency. For example, one can in theory employ a temporal learning strategy, which makes predictions based on previous trials and ignores transitions of contexts, to achieve accurate prediction in most trials except for when there is a change of CTD between blocks. To control for temporal learning and to determine whether participants learned the task demand probabilistically associated with each context...” (Page 6).

We apologize for the confusion about the comparison between the two models. As mentioned above, the temporal model was used to control for the confounding factor of temporal learning and to provide a more stringent test for CTD learning. Therefore, the relative contribution from the two models is not central to the test of our hypotheses. We revised the interpretation of the related behavioral results in the following manner:

“Given that both the contextual model and the temporal model were included in the same analysis to explain variance in the behavioral data, these latter findings indicate that, when temporal learning was accounted for, CTD was learned and additionally influenced task execution.” (Page 7, last sentence of behavioral results)

5) The formula “ $P_o(t) = P_o(t) + \alpha(T(t-1) - P_o(t-1))$ ” is not mathematically valid, as one of $P_o(t)$ in both hands should be replaced based on the model setup. For example, if the prediction is integrated, then the denotation should be $P_o(t-1)$ on the right hand of the formula. More details for each element will be also helpful. For example, if the P_o is over 0.5, does it predict the object task more, and vice versa for the face task (i.e., under 0.5)? Does the $T(t-1) - P_o(t-1)$ indicate PE? Then, shouldn't it be $|T(t-1) - P_o(t-1)|$ in the formula? Or PE was separately calculated? Also, it needs more details as to how the reinforcement learning model was differently built for the temporal model (using only trial sequence) vs. contextual model (using contextual information + trial sequence, in Methods, behavioral analysis, line 470). Was the only difference which trials were fed into the model for learning?

Response: We apologize for the typo. In the revised manuscript, the formula has been corrected to the following form:

$$P_o(t) = P_o(t-1) + \alpha(T(t-1) - P_o(t-1))$$

We also clarified the meaning of P_o in the following manner:

“Without loss of generality, we denote 0 and 1 for a face and an object trial, respectively. The reinforcement learning model learns to predict the task demand at trial t using the probability of performing an object task (denoted as $P_o(t)$, with $P_o(t)$ greater than 0.5 predicting higher likelihood of encountering an object task than face task), ...” (Page 22)

$T(t-1) - P_o(t-1)$ is the *signed prediction error*. We clarified that the behavioral analysis was based on $|T(t-1) - P_o(t-1)|$ in the following manner:

“Relatedly, trial-level (unsigned) prediction error, which was used in model-based behavioral analysis, was defined as $|T(t-1) - P_o(t-1)|$, or the absolute difference between T and P_o .” (Page 22)

Finally, we added more detail to the descriptions of the temporal and contextual learning models:

“The temporal model selectively used temporal information (i.e., ignored context changes), and consisted of only one reinforcement learner that is active throughout the experiment and learns task predictions from the sequence of trials. By contrast, the contextual model used a combination of temporal information and context to learn the CTD, such that there was one reinforcement learner for each of the four contexts. At each trial, only the learner associated with the present context was activated (i.e., no updating of P_o for the other three learners). The four contextual learners shared the same learning rate. Thus, at a given trial, the temporal model and the contextual model differ in: (1) the learning rate used, and (2) the learner updated (for the temporal model it is always the same learner, whereas in the contextual model it is the learner corresponding to the present context).” (Page 23)

6) Fig. 5D shows that context-context PS in hippocampus for the same CTD in different contexts predicts better CTD reinstatement in the prefrontal area, which means that less pattern separation for the different contexts sharing same CTD predicts better CTD

reinstatement. Isn't it the opposite from the prediction in the introduction (line 84)? The authors covered this opposite result in the discussion; however, it was still not justified enough how the inconsistent results could support the same argument.

Response: We apologize for the confusion in the introduction. We now clarified that pattern separation on contexts may have distinct effects on the memory of context (i.e., item memory) and the memory of context-CTD association (i.e., associative memory). The revised language now reads:

“Specifically, consider two pairs of contexts (dark and light dots in Fig. 1B, each dot representing one context), each associated with one CTD. Hippocampal coding of contexts within each pair may be separated to counter the interference between contexts caused by the shared CTD. On the other hand, pattern separation may hinder the context-cued retrieval of the associated CTD: relative to weaker hippocampal pattern separation (right panel of Fig. 1B), when pattern separation is strong (left panel of Fig. 1B) it is hypothesized that the cuing of one context is less likely to concurrently retrieve, or suffer interference from, the other context sharing the same CTD. In other words, strong pattern separation, while keeping contexts more distinct, fails to leverage the other context to boost retrieval and reinstatement of the shared CTD through recurrent retrieval^{3, 4}” (Page 4)

7) *In the model in the Fig. 1A, the process has one directional modulation from the hippocampus to behavioral outcome. I am curious if you can show the directional relationship for the CTD retrieval?*

Response: Thank you for this insightful comment. We added discussion on this issue to the revised manuscript, which reads:

“The direct modulation from the hippocampus to behavior is supported by two findings: (1) the relationship between hippocampal pattern differentiation in the Same CTD condition and CTD reinstatement in right dIPFC/frontopolar cortex (Fig. 5D), which modulated RT in trials later in the same block (Fig. 4C); and (2) the modulation of hippocampal activity at context onset on RT in trials later in the same block.” (Page 18, prior to conclusion paragraph)

8) *In Fig. 1B, it would be helpful if the figure shows how the separated representation predicts CTD learning.*

Response: Following the reviewer’s suggestion, we changed Fig. 1B to the following:

Stronger hippocampal separation of contexts sharing the same CTD may lead to more distinct neural coding of contexts (indicated by the distance between dots of the same color) and weaker facilitation (indicated by the thickness of the red arrows) from the other context to retrieve the associated CTD.

For convenience, the updated Figure 1 has been added to the end of this response letter (along with in the manuscript).

9) *(line 150) Was there no statistical significance for the accuracy in the temporal model?*

Response: We added the following result:

“The prediction error in the temporal model demonstrated a trend towards modulating accuracy (regression coefficient: 0.09 ± 0.05 ; $t_{32} = 1.81$; $P > 0.07$, one-sample t-test; $d = 0.31$).” (Page 7)

10) *The scale for the x-axis (PE) in the Figure 3 was missing. Did you group the PE in 5 scales or were there only 5 levels of PE? Why was the RT transformed to logged scale? This was never explained or justified. Also, the relationship in the Fig 3C seems to fit better for U-shape.*

Response: We now clarified in the figure legend that the x-axis represents data quintiles. The quintiles were for display purpose only, whereas the analyses were conducted at the trial level. We also clarified that the purpose of the log transform was to make the RT distribution more Gaussian (Page 23). Finally, a linear relationship between prediction error and (log) RT has been well established in previous studies using similar experimental designs^{5, 6}. Therefore, a linear model was used in the present study.

11) *What does “the context’s PS was computed in relation to out-of-run trials (line 179)” mean? Did you mean that you did not include PS within a run to remove auto-correlation?*

Response: Yes. In the Methods section (Page 27), we cited Cai et al (2018) to justify the exclusion of within-run pattern similarity scores.

12) *It would be easier to read the plots with significance symbols (asterisks) in Fig. 4C-F, middle column.*

Response: We added significance symbols to the middle column of Fig. 4C-F, based on the statistical results reported in Table 1. For convenience, the updated Figure 4 has been added to the end of this response letter (along with in the manuscript).

13) *In the Fig. 4B, right panel, was the data being used for the contrast still context-trial PS? Also, were the same task/other task same trials with congruent/incongruent on the left panel? And what is the scale of reinstatement at onset of context on the x-axis?*

Response: Yes, the context-trial pattern similarity was used for the CTD reinstatement-behavioral correlational analysis. We changed the text to congruent and incongruent for the table in the right panel of Fig. 4B. The x-axis represents the quintiles, which were for display purpose only (please also see above response to point 10).

14) *(line 183) Does “context and trial ...” means RSA between context and trial timepoints, same as “context-context RSA” for RSA in context timepoints between paired contexts? Then, using context-trial RSA would be more consistent.*

Response: Thank you for this suggestion. We now use the term ‘context-trial pattern similarity’ in the revised manuscript.

15) (line 274) What does “block-level” mean? Does it mean that the pattern was averaged within a block or the PS was conducted across blocks rather than within a block?

Response: We appreciate the request for further clarity here. The text, which now reads: “Here, context-context pattern similarity was calculated at the onset of the room for each block, again for Same Context, Same CTD, and Different CTD conditions (Fig. 5A).” (Page 13)

16) In Fig. 5C-D, it seems the term “modulation index (a.u.)” was not defined. Also, the scale on the x-axis in panel D were missing.

Response: We now defined the modulation index in the caption of Fig. 5C, which reads: “Group mean (\pm MSE) of the modulation index of each of the four predictors and the ROI univariate activity on the CTD reinstatement in each of the four frontoparietal ROIs. Modulation index is defined as the regression coefficient of the regressor (e.g., pattern similarity, univariate activity) on CTD reinstatement measured in the dlPFC/frontopolar ROI.”

We have clarified that the x-axes represent quintiles in the legend of Fig. 5D.

17) (line 500) Why were the post-error trials excluded?

Response: We added the following clarification to the revised manuscript: “Specifically, post-error trials are known to display ‘post-error slowing’, possibly due to a cautionary shift in response thresholds⁷, which represents a process that is not targeted in the model-based behavioral analysis or the retrieval of CTD analysis.” (Page 23).

18) (line 564) Was the separate GLM built for each trial or were all regressors with each regressor for each trial built in one GLM?

Response: We clarified this in the following manner: “A single GLM was constructed for each of the five runs.” (Page 26)

19) For pattern similarity analysis, what did you mean by normalized beta? Does it mean voxel-wise normalization of the beta estimate modeled on each trial? Also, did you use all voxels under each defined ROIs? Then, what was the number of voxels selected for each PS?

Response: We added further clarification, which reads: “For each ROI and each event, its activity pattern was quantified as a vector of multi-voxel normalized betas by dividing the original betas by the square root of the covariance matrix of the error terms from the GLM estimation⁸. All voxels in the ROIs were used in the calculation of pattern similarity.” (Page 26)

Regarding the number of voxels used in the ROIs, please see response below to Reviewer 3’s point 4.

20) *Was the context also modeled separately for the beta patterns? And given that it was presented for 7.5 seconds, was it modeled with boxcars?*

Response: We clarified this point in the revised manuscript in the following manner:

“To obtain event-level beta estimates for brain activity, each event (i.e., onset of building cue/building exterior and onset of room interior for each block, and onset of task cue for each trial) was represented by a single regressor of a hemodynamic function time-locked to the onset of the event. Each event was modeled using a stick function, because it was a priori unclear whether the learning and retrieval of CTD would last for the whole duration of stimulus presentation.” (Page 26)

21) For the Methods, while the preprocessing for the imaging data was described so detailed (maybe too specific), other analyses were not described enough.

Response: We apologize for the lack of detail in the analysis methods. In addition to the details added based on your and other reviewers’ comments, we included more information to help clarify the methods. To facilitate replication, the analysis scripts were also made available via the link in the Data Availability Statement.

Reviewer #2

1) *We found it difficult to distill the primary claim of the paper. It appears to be that hippocampal activity, indicated by pattern separability, indexes reinstatement of “contextual task demands” in DLPFC, which subsequently modulate behavior. The directionality and causal nature of the relationship appear to be central to the claim - the manuscript is peppered with language such as that “hippocampal representations of context modulate proactive retrieval” (Line 38) and “CTD depends on the hippocampus” (Line 73).*

If this is the main finding, it leads to two concerns:

- First, it’s not clear what this finding would add to previous work, including work from the author’s own lab. Specifically, it has been shown that contextual task demands (stimulus-action tendencies) are reinstated by hippocampus on the basis of informative cues (Hindy & Turk-Browne 2016, 2019), and that contextual task demands can be decoded from PFC activity patterns preceding performance of the task, and used to predict performance on the subsequent task (Waskom et al 2014, 2016, 2017). Moreover, prior work has shown task-dependent tuning of sensory processing (Tajima et al., 2017) and sensorimotor processing can be tuned according to the anticipated demands (Muhle-Karbe et al., 2017). Given these findings, it would be helpful if the authors could pull reinstatement results from visual ROIs (e.g., face area and object area and/or V1) as control regions compared to the DLPFC to rule out perceptual facilitation and 2) include a discussion explicating how their neural findings are similar and/or distinct from, e.g., the Hindy findings.

Response: We thank the reviewer for this guidance. Following their input, we performed the reinstatement analysis on visual cortex and now report the findings:

“To examine the reinstatement of CTD in the visual cortex, we tested the interaction between context and congruency (Fig. 4B, left panel) in 34 visual ROIs (defined by the multi-modal cortical parcellation from the Human Connectome Project; major assignment IDs: 1, 2, 3,

and 4; ROI size = 269 ± 52 voxels; range = 49–1183)⁹. After correction for multiple comparisons, only the left V8 ROI exhibited a significant context \times congruency interaction effect ($t_{32} = 3.69$, $P < 0.001$). However, the reinstatement of CTD at the beginning of a block in left V8 did not modulate behavior later in the block ($t_{32} = 0.80$, $P = 0.43$, analysis conducted using the contrast shown in the right panel of Fig. 4B). Given that the right dlPFC/frontopolar ROI exhibited both a context \times congruency interaction and modulation of CTD reinstatement on behavior, the lack of visual areas showing both effects suggests that the prefrontal reinstatement of CTD is not solely explained by perceptual facilitation.” (Supplementary Note 2)

We further discussed how our major findings relate to prior work and substantively advance understanding of the acquisition and expression of learned context-control state associations. First, we note that compared to Hindy et al.’s (2016, 2019) focus on pattern completion, the main conclusion of our work concerns the role of pattern separation in building representations:

“Multi-variate hippocampal activity patterns¹⁰ and connectivity¹¹ provide additional support for the mechanism of pattern completion during cued retrieval of stimulus-action sequences. To further study the interaction between CTD reinstatement and hippocampal mnemonic mechanisms, we examined the relationship between hippocampal pattern separation and CTD reinstatement in dlPFC.” (Page 18)

Compared to Waskom et al (2014) and other related work focused on the neural representation of task-set in frontoparietal networks, our work targets the interaction between the hippocampus and the PFC during the retrieval of associated task demand. We first clarified this using the sentence below:

“Consistent with previous findings showing instantiation of task-set during task in PFC¹²⁻²⁰, the context-trial pattern similarity analyses revealed frontoparietal foci in which reinstatement of CTD was observed, including right dlPFC (Fig. 4C, left column). ... We then investigated the relationship between CTD reinstatement in dlPFC and hippocampal activity at the onset of the spatial context.” (Page 17)

We then explained that Waskom et al (2017) and the present work examine different sources of task demand prediction:

“One reinforcement learning model (contextual model) simulated the learning of CTD²¹, and a second reinforcement learning model (temporal model) simulated context-insensitive learning of task demand through temporal information^{5, 6}.” (Page 7)

2a) - Second, and more importantly, it is not clear that this chain of events is supported by the analyses presented. Specifically, the claim would require:

a. That hippocampal reinstatement precede dlPFC reinstatement. However, both activity patterns are decoded during the same time interval. No evidence in support of the necessary ordering is presented - and this would be difficult to do in fMRI. In fact, there is evidence in the literature of the opposite order of cue-evoked processing, though not in the exact same task (Hung et al 2013).

Response: We thank the reviewer for this insightful comment. We agree that testing directionality of modulation is challenging in fMRI studies. In the revised manuscript, we now explicitly acknowledge that our conclusions about directionality await further direct assessment using higher temporal resolution methods. Moreover, we conducted a control analysis to explore whether dlPFC reinstatement might stem from an alternative source:

“One alternative interpretation of the results is that hippocampal pattern separation is mediated by CTD reinstatement in the dlPFC. If this were true, one might predict that the dlPFC reinstatement would be modulated by other learning systems. We tested this prediction by repeating the analyses in the striatum (defined using the labels caudate, putamen and accumbens areas from FreeSurfer). CTD reinstatement in dlPFC was not significantly modulated by striatal univariate activity nor by striatal context-context pattern similarity in any of the Same Context, Same CTD, or Different CTD conditions (all P s > 0.33). This result lends indirect support to the argument that CTD reinstatement in the dlPFC is modulated by hippocampal activity and pattern differentiation.” (Supplementary Note 7)

2b) *Excluding the possibility that correlations between hippocampal activity, DLPFC activity, and behavior are driven by a common, unobserved, cause. Here, evidence for this is that the behavioral model predictions are relatively improved (but see below for technical questions regarding this analysis) by the addition of HC and DLPFC reactivation measures. This does not exclude the possibility that all three are reflective of variability in learning performance that is not captured by the model, but which originates from neural processing unrelated to these regions. The relationship between activity and behavior is itself unconvincing, because functions commonly ascribed to these regions are also reflective of underlying causes that could occur in parallel to the behavioral variability of interest (e.g. stimulus surprise or novelty detection, pre-fetching of upcoming stimuli or stimulus-action associations, working memory contents or general load).*

Response: We appreciate this point. First, we note that the above analysis suggests that the striatum is not a common modulator. Second, we performed the following additional control analyses to search for potential common modulators in the cortex:

“An additional possibility is that both the dlPFC reinstatement of CTD and the hippocampal context-context pattern similarity in the Same CTD condition are mediated by a common modulator. To test this hypothesis, we searched for potential common modulators in the 359 cortical ROIs (excluding right BA 9-46d) defined by the multi-modal cortical parcellation from the Human Connectome Project. Specifically, for each ROI, its univariate activity (which may vary with general load, surprise, or novelty) and context-trial pattern similarity in the Same Context (reflecting stimulus-level content representation) and the Same CTD (reflecting task demand-level representation) were used to explain variance of both the dlPFC reinstatement of CTD and the hippocampal context-context pattern similarity in the Same CTD condition at the block level. Individual modulation strength was then tested against 0 at the group level using one-sample t-test. Multiple comparisons were leniently corrected using FDR within each ROI, in order to reduce type II error. A common modulator would show significant modulation in at least one of its three measures above on both dlPFC reinstatement of CTD and hippocampal pattern differentiation. However, in contrast to a common modulator hypothesis, no ROI met the criteria. However, as this is a null result, interpretative caution is warranted. Future studies are encouraged to further examine the directionality of the relationship between dlPFC reinstatement of CTD and hippocampal activity patterns at retrieval.” (Supplementary Note 7)

3) *A very intriguing claim, somewhat obscured in the manuscript (Discussion, Lines 398-402) is that pattern separation entails a tradeoff against the cost of reduced transfer learning between instances of the same task that were preceded by different cues. This would be an interesting result, and should be highlighted earlier on in the paper. However, it needs further*

substantiation, linking pattern similarity measures to subsequent performance improvements (or decrements). For instance: is it the case that pattern similarity following cue A at timepoint t predicts improved performance (on match trials) on that same task at later timepoints $t+x$ (and worsened performance on mismatch trials), following Same-CTD cue B?

Response: We appreciate this insightful comment. We did not find a significant relationship between hippocampal pattern separation and behavioral performance at the block level (see last paragraph of Results). The analysis proposed by the reviewer is intriguing, yet the expected effect may be too weak to be detected given that performance at later time points $t+x$ will also (and probably more strongly) be affected by Same CTD context-context pattern similarity when context B was presented. Therefore, to explore the reviewer's point, we performed and report an alternative analysis that provides some purchase on the issue:

"The argument that hippocampal pattern separation balances integration and separation would predict that stronger pattern separation will lead to stronger facilitation in behavior. We tested this prediction by correlating the Same CTD context-context pattern similarity in the hippocampus with the behavioral modulation of CTD shown in Fig. 3 (separately for accuracy and RT) across participants. Consistent with this prediction, we observed a significant correlation ($r = -0.39$, $P = 0.025$, Supplementary Fig. 6), such that, contextual prediction errors impact accuracy more (indicating stronger behavioral influence of CTD) when the pattern similarity scores are lower (indicating stronger separation)." (Page 19)

Supplementary Figure 6. Individual modulation of contextual prediction error on accuracy (lower value indicates stronger modulation), plotted as a function of the Same CTD context-context pattern similarity in the hippocampus (lower value indicates stronger separation). Dashed line represents linear trend line.

4a) There are several places in the manuscript where the authors use multiple regression to support the claim that covariate B accounts for variance unexplained by covariate A - e.g. in the comparison of the temporal and contextual models, and in the However, this approach is only valid when the covariates are perfectly orthogonal. Can the authors report the correlation

between regressors, and whether the results hold if the regressors are orthogonalized against each other, and/or the regression of each is run on the residuals after regression on the other?

Response: We thank the reviewer for this thoughtful comment; we added the following control analysis:

“Given that temporal prediction error and contextual prediction error are correlated ($r = 0.44 \pm 0.04$, range: 0.10-0.87), we replicated the previous behavioral analyses using residuals of contextual prediction error after regression on the temporal prediction error, and obtained qualitatively similar results for both accuracy (regression coefficient: -0.19 ± 0.04 ; $t_{32} = -4.52$; $P < 0.001$, one-sample t-test; $d = 0.79$) and RT (regression coefficient: 0.01 ± 0.004 ; $t_{32} = 2.64$; $P = 0.01$, one-sample t-test; $d = 0.46$). In addition, after regressing shared variance with contextual prediction error, temporal prediction error still exhibited significant modulation on trial-level RT (regression coefficient: 0.03 ± 0.004 ; $t_{32} = 9.29$; $P < 0.001$, one-sample t-test; $d = 1.62$).” (Supplementary Note 1)

4b) *In the case of the model comparison, an alternative approach could be to combine the two models into a single model, for instance by using a linear weighting parameter (e.g. w *temporal + $(1-w)$ *contextual). This would allow them to compete for prediction of behavior on equal terms and permit statistical analysis commensurate with the regression weights reported, while still providing separable estimates of trial-by-trial activity.*

Response: Following the reviewer’s suggestion, we added a new analysis:

“To quantitatively assess the joint contribution of temporal and contextual predictions of task demand to behavior, we designed an additional model, which combines both predictions and takes the form:

$$P_o(t) = w * P_o^c(t) + (1-w) * P_o^t(t)$$

Where P_o is the joint prediction of task demand, which is a weighted sum of P_o^c and P_o^t (denoting the contextual and temporal predictions of task demand [see Methods: Behavioral Analysis], respectively). The model includes three free parameters (the weight w and the respective learning rates for P_o^c and P_o^t), which were determined using a grid search (w range: 0-1, step size = 0.01; learning rate range: 0.01-0.99, step size = 0.01) that maximizes the variance explained in the trial-wise RT data (see Methods: Behavioral Analysis). At the group level, the weight was significantly lower than 0.5 ($w = 0.30 \pm 0.05$, $t_{32} = 4.20$; $P < 0.001$, one-sample t-test; $d = 0.73$), indicating that subjects rely more on temporal than contextual predictions. Furthermore, the joint prediction error significantly modulated accuracy (regression coefficient: -0.24 ± 0.04 ; $t_{32} = -5.72$; $P < 0.001$, one-sample t-test; $d = 1.00$) and RT (regression coefficient: 0.04 ± 0.003 ; $t_{32} = 13.53$; $P < 0.001$, one-sample t-test; $d = 2.36$).” (Supplementary Note 8)

We would like to further note that the purpose of modeling temporal and contextual predictions separately is to explicitly tease apart the behavioral modulation of temporal prediction and to be able to test the learning of the CTD directly. For this purpose, we decided to keep the two learners modeled separately in the main text.

4c) *It seems as though a strong test of the author’s hypothesis might be that the pattern similarity measure predicts the value of this weighting parameter, across trials or runs. There are many reasons that this wouldn’t work, but it may be worth investigating.*

Response: We appreciate this insightful prediction. We agree with the reviewer that increase in pattern separation is predicted to lead to stronger modulation of CTD on behavior. This is now tested at the individual level, in the response to point 3 above. We also tested the correlation between the weighting parameter and Same CTD context-context pattern similarity across subjects, which was not significant ($r = 0.02$, $P > 0.91$). We speculate that this null result is due to the weighting parameter being unable to account for individual differences in the modulation of prediction error on behavior.

4d) *In general, the authors claims would be better supported if they could show that some neurally-derived measure improves the predictive power of the behavioral model.*

Response: We appreciate the reviewer's point, while also noting that our view is that there should be some shared variance explained by the neural measure and behavioral models. To test whether the neural measure explains behavioral data that is above and beyond what is accounted for in the behavioral models, we now report the following:

"We repeated the analysis above using both contextual prediction error and temporal prediction error as covariates, and observed a marginally significant modulation of dlPFC/frontopolar CTD reinstatement on RT (regression coefficient: -0.0040 ± 0.0022 ; $t_{32} = -1.81$; $P = 0.079$, one-sample t-test; $d = 0.32$). This finding provides initial support for the claim that the neural measure of CTD reinstatement explains behavioral data above and beyond what is accounted for in the behavioral models." (Page 11)

4e) *Is it the case that the contextual model fits behavior "better" than the temporal model? Or vice-versa? If the claim is that the contextual model predictions underlie the neural activity, and on this basis the temporal model is excluded from MRI analysis, it seems important to justify this exclusion - either by showing a better fit to behavior, or by identifying separable neural correlates.*

Response: As noted above in the response to point 4b, the participants appear to rely more on temporal than contextual prediction of task demand. That said, the goal of the behavioral analysis is to test whether the participants also learned the contextual task demand and used it to guide behavior, rather than to test if the contextual model has a stronger behavioral modulation than the temporal model. The behavioral analyses indeed reveal that the prediction error of the contextual model significantly mediates both accuracy and RT. Again, we further clarified the logic of these analyses in the manuscript.

In addition, we note that the fMRI analyses concerned activity patterns at the onset of the contexts and did not rely on parametric output from either model. Thus, we expect that conducting model comparison or running fMRI analysis on the temporal model would not change the main findings and the conclusion.

4f) *Can the authors present a comparison of the contextual (and temporal) models against a model that allows learning rate to vary between contexts? This might have been a minor comment, but for the above query about whether neural responding and behavior may be both driven by learning variability not captured by the model.*

Response: The goal of using contextual and temporal models is to look for behavioral evidence of the learning of CTD, while accounting for temporal learning of task demand. This

goal is achieved using the current contextual and temporal models. According to Behrens et al (2007) and Jiang et al (2015), the learning model with a changing learning rate is more effective in an unstable environment. In the context of this study, an unstable environment would mean (1) the CTD varies over time and/or (2) the rate of the variation shifts over time. Neither of these two criteria applies to the experimental design in this study, because the CTDs were constant. Thus, we think that using a model with a changing learning rate will not qualitatively change the behavioral results. We now provide a brief justification for adopting a fixed learning rate in the manuscript:

“Recent studies have shown the benefit of adopting self-adjusting learning rates in learning models^{22, 23}. The benefit is more pronounced in changing environments (e.g., when CTD changes over time in the context of the present experimental design). Given that the CTD in the experimental design stayed constant, we chose a simple fixed learning rate.” (Page 24)

5) *We suggest that the authors reconsider using “PS” to refer to pattern similarity, as the manuscript also relies on the term “pattern separation” or “pattern separability” at several points. This confused us in more than one instance. Perhaps “PSim” might be a better abbreviation?*

Response: Following this suggestion and related comment from Reviewer 1, we now use the full term “pattern similarity” throughout the manuscript.

6) *The labels of the regions in each MRI image are a bit obscure, especially in contrast to the more simplified names used elsewhere in the manuscript. Can the authors please choose descriptive labels and use them consistently throughout the manuscript?*

Response: We apologize for the obscurity. In the revised manuscript, we use both the descriptive labels and the labels from the Human Connectome Project, in order to be intuitive and precise. Please refer to the updated Fig. 4 and Fig. 5 (also attached to the end of this response letter for convenience).

7) *The use of a grid-fitting approach is strange. While we don’t necessarily suspect that this introduced any confounds, why don’t the authors employ a more standard optimization approach?*

Response: We appreciate the reviewer’s concern. Given the low number of free parameters, we chose to use a grid search to cover the full search space, as compared to randomly choosing starting points (which may be trapped in local minima) as in most optimization procedures. We now briefly comment on this logic in the Methods (Page 23).

8) *The operational definition of pattern similarity is intriguing and may obscure some issues. Specifically, similarity is defined across runs (Page 9, Lines 178-179), which might lead to a confound with learning, as similarity should only decrease across time. Is there a common reference point against which pattern similarity can be measured? Or could the authors instead employ a dissimilarity measure (e.g. against patterns evoked by the other cues within the same run)?*

Response: We thank the reviewer for raising this important issue. We tested whether there was systematic temporal change of pattern similarity measures in the following manner:

“We examined whether there was a systematic temporal change in context-context pattern similarity by calculating the correlation between block-level pattern similarity measures (each of the Same context, Same CTD and Different conditions) and time (each of block ID and the logarithm of block ID) for each participant. One sample t-tests against 0 at the group level did not reveal any significant temporal change that was consistent across participants (all $P_s > 0.56$).” (Supplementary Note 5)

We decided not to calculate within-run pattern similarity due to the potential auto-correlation confound that may be introduced²⁴.

9) Page 20, Line 474: *The left side of the equation should read $P_o(t+1)$.*

Response: We apologize for this typo. It has been fixed in the revised manuscript.

10) Page 24, Line 578: *Activity was extracted from bilateral hippocampus, were the final effects with DLPFC driven by the right hippocampus?*

Response: Following this guidance, we tested this possibility and added the following:

“When repeated separately for the left and right hippocampus, the regression coefficient for the Same CTD context-context pattern similarity did not significantly differ between the two hemispheres (left: 0.037 ± 0.038 ; right: 0.044 ± 0.032 ; $t_{32} = 0.12$; $P = 0.9$; paired t-test).” (Page 14)

11) Is the “contextual task demand” distinct from a response bias? That is, the authors show that RT and accuracy are correlated with the “prediction error” derived from their models, and suggest that this is indicative of ongoing conflict between task performance and reinstated task demands. However, this prediction error is exactly one minus the predicted probability of the task, which has associated with it a pair of responses distinct from those of the other task. Does this effect persist throughout all eight trials in each room? Or is it largely found in the first few, and then decays, as might be expected of a response bias?

Response: The task demand includes response mapping, thus a response bias is assumed to be part (but not all) of the contextual task demand effect. We examined the univariate activity in motor and premotor cortex at the onset of the context (i.e., the room) but did not find strong evidence of proactive activity favoring the task predicted by the CTD (please see the response below to Reviewer 3’s point 3). We also added the following analysis:

“Furthermore, to test whether motor and premotor reinstatement of CTD in the activity patterns at the context onset affects behavior later in the block, we tested the CTD reinstatement modulation on RT (using the contrast shown in the right panel of Fig. 4B and an identical procedure to that used with the frontoparietal ROIs) in each of the aforementioned 24 ROIs. None of the ROIs showed significant behavioral modulation (all $P_s > 0.09$). Taken together, we did not observe strong evidence for response bias at the context onset.” (Supplementary Note 4)

Reviewer #3

1) Individual subject data - Throughout the paper the authors report quintiles for their effects. This approach nicely illustrates their main findings, however, it would be useful to show the individual subject data for these analyses. This is a within subjects design and showing the individual subject data is important for understanding variability in the effect. Furthermore, the authors state the first run of scanning was thrown out because subjects had not yet learned the task. It would be nice to include learning curves for individual subjects to illustrate this effect.

Response: Following the reviewer's comment, we added individual data for each of the quintile plots. The added figures are attached below:

Supplementary Figure 1. Individual quintile data corresponding to Fig. 3B (left) and 3C (right).

L dlPFC/frontopolar
(BA 9-46d)

L superior frontal
(BA i6-8)

L inferior frontal
junction (IFJp)

L superior parietal
(BA 7PL)

Supplementary Figure 2. Individual quintile data corresponding to Fig. 4C-F.

Supplementary Figure 5. Individual quintile data corresponding to Fig. 5D.

The exclusion of Run 1 was based on the consideration that: (1) it was the beginning of the learning of CTD; (2) each context only has 2 blocks in Run 1, providing limited exposure to the participants; and (3) the main goal of the present study is the retrieval of CTD, thus we discarded Run 1 to exclude data before the CTD was learned. The learning curve based on expected vs. unexpected tasks may be confounded by temporal learning. To selectively assess the learning of CTD, we compared the hippocampal representation of contexts in the Same CTD and Different CTD conditions and examined how the difference between the two conditions evolved over time. As shown above in the response to Reviewer 1's point 3a, pattern differentiation was absent in Run 1 but emerged subsequently, suggesting that Run 1 reflects performance (and associated neural responses) that falls prior to the completion of CTD learning.

2) Medial ROIs - mPFC has been shown to be important for context dependent memory retrieval and certain aspects of generalization behavior. It could be of interest for the authors to examine pattern similarity in this regions for the Same CTD condition and how that relates the dlPFC effects reported in the manuscript.

Response: Following this insightful comment, we added the following analysis:

“Given the importance of the medial prefrontal cortex in context-dependent memory retrieval and generalization, we compared the context-context pattern similarity in the Same CTD condition with the Different CTD condition in each of the 32 ROIs in the ‘ACC and medial prefrontal cortex’ category (ROI size = 166 ± 30 voxels; range = 61–446) of the multi-modal cortical parcellation from the Human Connectome Project. Only the left pOFC area exhibited a significant reduction of context-context pattern similarity in the Same CTD than the Different CTD condition, and this was true only when using an uncorrected alpha-level (difference: 0.0035 ± 0.0017 , $t_{32} = -2.11$; uncorrected $P = 0.043$, paired t-test; $d = 0.37$). We then tested the modulation of the Same CTD condition context-context pattern similarity on the CTD reinstatement in the right dlPFC (see Fig. 5C) for each of the ROIs, but did not find any ROI showing the positive modulation that was observed for the hippocampus (all $P > 0.051$). These results suggest a limited involvement of the medial prefrontal cortex in the retrieval of the CTD.” (Supplementary Note 3)

3) It may be of interest to the authors to examine the relationship between univariate activity in motor regions and the dlPFC same CTD pattern similarity effect. Given that the task sets are paired with different handed button responses, we might expect an effect where when subjects enter a context you get modulation of activity in motor regions in a task dependent way. This analysis could expand on and further support the hypothesis that dlPFC is contributing to retrieval of task-sets via a suppressive mechanism.

Response: Following the reviewer's suggestion, we tested the univariate activity in the motor and premotor areas. The related text reads:

“To examine whether and how the retrieval of the CTD mediates motor planning and response preparation, we tested the univariate activity at the onset of the context in each of the 12 premotor, somatosensory, and motor cortex ROIs (major assignment IDs 6 and 8 in the multi-modal cortical parcellation from the Human Connectome Project; collapsed across left and right hemispheres; ROI size = 371 ± 32 voxels; range = 71–1106). The univariate activity

level was defined as the ROI-mean of the multi-voxel normalized betas⁸, in order to be consistent with the pattern similarity analyses. Given that participants used different hands for the two tasks in our experimental design, retrieval of the CTD predicts increased univariate activity in motor/premotor areas contralateral to the hand used for the task predicted by the CTD, as compared to the ipsilateral hand. Although we observed such a pattern in the L_1 area when using an uncorrected alpha-level (contralateral - ipsilateral: 0.065 ± 0.027 , $t_{32} = 2.43$; uncorrected $P = 0.021$, paired t-test; $d = 0.42$), no ROI survived the FDR correction.” (Supplementary Note 4)

4) *You have chosen to use the human connectome project parcellation for your regions of interest in this paper. Some of these ROIs are quite small. It would be useful to report the number of voxels in each ROI on average.*

Response: We now report the mean \pm MSE and the range of ROI size in the revised manuscript. The numbers are summarized in the table below:

ROI	Mean \pm MSE (voxels)	Range (voxels)
Frontoparietal (main text)	198 \pm 16	51–511
Medial PFC (response to point 2)	166 \pm 30	61–446
Motor and premotor (response to point 3)	371 \pm 32	71–1106
Visual areas (response to R2’s point 1)	269 \pm 52	49–1183

5) *Regarding the above point, in the manuscript the authors mention 4 ROIs that were used, but the reports lateralized effects. Please be clear that lateralized ROIs were used in the main text.*

Response: Following the reviewer’s suggestion, we now report the hemisphere of the four PFC ROIs in the revised manuscript. In the methods section, we also clarified that lateralized ROIs were used.

6) *Table 1 shows four prefrontal ROIs but statistics reported in the main text report lateralized effects. Please make sure these statistics are aligned.*

Response: We added the hemisphere to each of the four PFC ROIs. The labels in figures were also changed accordingly. Table 1 now reads (hemispheres highlighted in red font):

	R BA 9-46d	L BA i6-8	L IFJP	L BA 7PL
Same Context + Same CTD	0.0026 \pm 0.0007	0.0021 \pm 0.0007	0.0028 \pm 0.0011	0.0023 \pm 0.0012
	3.75***	3.04**	2.61*	1.92
Same CTD	0.0033 \pm 0.0014	0.0020 \pm 0.0011	0.0037 \pm 0.0014	0.0021 \pm 0.0014
	3.65***	1.85	2.59*	1.57
Different CTD	-0.0012 \pm 0.0009	-0.0017 \pm 0.0008	-0.0036 \pm 0.0011	-0.0046 \pm 0.0010
	-1.38	-2.16*	-3.51**	-4.54***

Table 1. Summary statistics of the context-trial pattern similarity congruency effects in the four cortical ROIs showing significant context \times congruency interactions in context-trial pattern similarity. For each condition, the top and

bottom rows show the group mean \pm MSE and the t-value (DOF = 32), respectively *: $p < 0.05$; **: $p < 0.01$; ***: $p < 0.001$.

7) *Correction for multiple comparisons are applied in some parts of the manuscript but not all. Please be explicit about where they are applied by including corrected or uncorrected next to the reported p-value.*

Response: Because we used FDR correction to control for multiple comparisons, the uncorrected p-values are not intuitive to be converted into corrected p-values. As an alternative approach to addressing this point, we now state the following:

“Unless otherwise specified, all reported P-values were uncorrected. We also report whether the reported results survived FDR correction.” (Page 27)

In short, the right dlPFC/frontopolar ROI survived all multiple comparisons when controlled for the number of ROIs tested and/or the number of modulators tested. The hippocampal context-context pattern similarity results also survived FDR. Therefore, FDR correction did not change the main findings of the present study.

8) *It is important for reproducibility that the authors report the software and version used for statistical analyses performed on the pattern similarity data.*

Response: We thank the reviewer for this guidance. We now report that:

“All pattern similarity analyses and statistical tests were conducted using Matlab 2017a” (Page 27). As reported in the Data Availability Statement, related analysis scripts are shared on Github (<https://github.com/JiefengJiang/CTD>).

9) *Presumably the star and the bracket in Figure 5b reflects a significant between-condition difference, but I didn't see this stated in the caption*

Response: We now state the statistical significance in the caption of Fig. 5b, which reads:

“(B) Visualization of the hippocampus ROI (top, in red) and the group mean (\pm MSE) of the three conditions of hippocampal context-context pattern similarity (bottom). At the group level, pattern similarity in the Same CTD condition was significantly lower than the Different CTD condition.” (Page 16).

10) *I am not sure that “pattern separation” is the best way to describe the hippocampal results. I realize that this is true in the literal sense (i.e., the hypotheses concern differences in fMRI pattern similarity), the authors are referring to theories suggesting that the hippocampus assigns sparse, minimally overlapping representations to similar inputs. While it is true that the hippocampus differentiates between the Same context trial pairs and other conditions, the exaggerated reduction of pattern similarity for the Same CTD trials does not necessarily fit with traditional theories of pattern separation (e.g., Guzowski et al., 2004). By describing this effect in terms of pattern separation, I think the authors are making the data sound more prosaic than they really are—the finding of exaggerated differentiation for similar contexts/items is well-documented (e.g., work by Brice Kuhl's lab) and it is a finding that isn't easily explained by sparse coding alone. There are theories that try to account for this kind of effect (e.g., Ritvo et al., 2019) that may be worthy of consideration.*

Response: We thank the reviewer for this insightful comment. In the revised manuscript, we discussed the difference between the traditional conceptualization of pattern separation and pattern differentiation in the following manner:

“Distinct from the pattern separation theory that different memory traces are encoded in sparse neural codes, this effect assumes partially overlapping neural codes for overlapping experiences. Through the course of learning, the shared neural representations are pruned (possibly due to a pruning process following weak activation of the overlapping portion of the neural codes²⁵) and lead to more differentiated codes for the experiences². We first replicated this effect...” (Page 18)

11) *Related to above, the authors cite McClelland, McNaughton, and O’Reilly in reference to pattern separation. I may be misremembering, but I did not think that paper emphasized pattern separation and pattern completion, I thought this was emphasized in O’Reilly’s later, more biologically-based models (as well as work by Marr, Ed Rolls, etc.).*

Response: Thank you for catching this citation error. We now cited McClelland and O’Reilly (1994), *Hippocampus* as a reference to pattern separation.

12) *I also felt that this statement was overly speculative given what the authors actually found in the study: “The reduction in hippocampal PS in the Same CTD condition may facilitate the separation of and memory for individual contexts, as suggested by prior work documenting that greater hippocampal pattern distinctiveness is associated with better subsequent memory performance at the item level”*

Response: We apologize for this confusion. Given that this statement is not central to the present study, we have removed it from the revised manuscript.

References

1. Cohen, N.J. & Eichenbaum, H. *Memory, amnesia, and the hippocampal system* (MIT Press, Cambridge, Mass., 1993).
2. Favila, S.E., Chanals, A.J. & Kuhl, B.A. Experience-dependent hippocampal pattern differentiation prevents interference during subsequent learning. *Nat Commun* **7**, 11066 (2016).
3. Kumaran, D. & McClelland, J.L. Generalization through the recurrent interaction of episodic memories: a model of the hippocampal system. *Psychological review* **119**, 573-616 (2012).
4. Koster, R., *et al.* Big-Loop Recurrence within the Hippocampal System Supports Integration of Information across Episodes. *Neuron* **99**, 1342-1354 e1346 (2018).
5. Jiang, J., Wagner, A.D. & Egner, T. Integrated externally and internally generated task predictions jointly guide cognitive control in prefrontal cortex. *eLife* **7** (2018).
6. Waskom, M.L., Frank, M.C. & Wagner, A.D. Adaptive Engagement of Cognitive Control in Context-Dependent Decision Making. *Cerebral cortex* **27**, 1270-1284 (2017).
7. Danielmeier, C. & Ullsperger, M. Post-error adjustments. *Frontiers in psychology* **2**, 233 (2011).
8. Walther, A., *et al.* Reliability of dissimilarity measures for multi-voxel pattern analysis. *NeuroImage* **137**, 188-200 (2016).
9. Glasser, M.F., *et al.* A multi-modal parcellation of human cerebral cortex. *Nature* **536**, 171-178 (2016).
10. Hindy, N.C., Ng, F.Y. & Turk-Browne, N.B. Linking pattern completion in the hippocampus to predictive coding in visual cortex. *Nature neuroscience* **19**, 665-667 (2016).

11. Hindy, N.C., Avery, E.W. & Turk-Browne, N.B. Hippocampal-neocortical interactions sharpen over time for predictive actions. *Nat Commun* **10**, 3989 (2019).
12. Dobbins, I.G., Rice, H.J., Wagner, A.D. & Schacter, D.L. Memory orientation and success: separable neurocognitive components underlying episodic recognition. *Neuropsychologia* **41**, 318-333 (2003).
13. Stokes, M.G., *et al.* Dynamic coding for cognitive control in prefrontal cortex. *Neuron* **78**, 364-375 (2013).
14. Sigala, N., Kusunoki, M., Nimmo-Smith, I., Gaffan, D. & Duncan, J. Hierarchical coding for sequential task events in the monkey prefrontal cortex. *Proceedings of the National Academy of Sciences of the United States of America* **105**, 11969-11974 (2008).
15. Schuck, N.W., Cai, M.B., Wilson, R.C. & Niv, Y. Human Orbitofrontal Cortex Represents a Cognitive Map of State Space. *Neuron* **91**, 1402-1412 (2016).
16. Collins, A.G., Cavanagh, J.F. & Frank, M.J. Human EEG uncovers latent generalizable rule structure during learning. *The Journal of neuroscience : the official journal of the Society for Neuroscience* **34**, 4677-4685 (2014).
17. Haynes, J.D., *et al.* Reading hidden intentions in the human brain. *Current biology : CB* **17**, 323-328 (2007).
18. Waskom, M.L., Kumaran, D., Gordon, A.M., Rissman, J. & Wagner, A.D. Frontoparietal representations of task context support the flexible control of goal-directed cognition. *The Journal of neuroscience : the official journal of the Society for Neuroscience* **34**, 10743-10755 (2014).
19. Wisniewski, D., Reverberi, C., Momennejad, I., Kahnt, T. & Haynes, J.D. The Role of the Parietal Cortex in the Representation of Task-Reward Associations. *The Journal of neuroscience : the official journal of the Society for Neuroscience* **35**, 12355-12365 (2015).
20. Dobbins, I.G. & Wagner, A.D. Domain-general and domain-sensitive prefrontal mechanisms for recollecting events and detecting novelty. *Cerebral cortex* **15**, 1768-1778 (2005).
21. Chiu, Y.C., Jiang, J. & Egnér, T. The Caudate Nucleus Mediates Learning of Stimulus-Control State Associations. *The Journal of neuroscience : the official journal of the Society for Neuroscience* **37**, 1028-1038 (2017).
22. Behrens, T.E., Woolrich, M.W., Walton, M.E. & Rushworth, M.F. Learning the value of information in an uncertain world. *Nature neuroscience* **10**, 1214-1221 (2007).
23. Jiang, J., Beck, J., Heller, K. & Egnér, T. An insula-frontostriatal network mediates flexible cognitive control by adaptively predicting changing control demands. *Nat Commun* **6**, 8165 (2015).
24. Cai, M.B., Schuck, N.W., Pillow, J.W. & Niv, Y. Representational structure or task structure? Bias in neural representational similarity analysis and a Bayesian method for reducing bias. *PLoS computational biology* **15**, e1006299 (2019).
25. Ritvo, V.J.H., Turk-Browne, N.B. & Norman, K.A. Nonmonotonic Plasticity: How Memory Retrieval Drives Learning. *Trends in cognitive sciences* **23**, 726-742 (2019).

Updated figures

Figure 1. Study overview. (A) Proposed cognitive processes underlying the reinstatement of learned CTD. Blue text indicates the figures where relevant results are shown. (B) Stronger hippocampal separation of contexts sharing the same CTD leads to more distinct neural coding of contexts (indicated by the distance between dots of the same color) and weaker facilitation (indicated by the thickness of the red arrows) from the other context to retrieve the associated CTD.

B Context \times congruency interaction

	Same context	Same CTD	Different CTD
Congruent	1	1	-2
Incongruent	-1	-1	2

Reinstatement at onset of context

	Same context	Same CTD	Different CTD
Congruent	1	1	0
Incongruent	-1	-1	0

C L dlPFC/frontopolar (BA 9-46d)

D L superior frontal (BA i6-8)

E L inferior frontal junction (IFJp)

F L superior parietal (BA 7PL)

Figure 4. Frontoparietal reinstatement of CTD. (A) Four examples (grouped by blue background) of how the CTD of context A (top row) and the required task and the CTD of a trial in context B (bottom row) define the experimental condition for testing the reinstatement of CTD (text label below the background). (B) Left: linear contrast testing the reinstatement of CTD. To account for different frequencies of trial types and to make the contrast orthogonal to main effects, the weight for Different CTD conditions was twice the weight for Same Context and Same CTD conditions. Right: Contrast used to test the modulation of CTD reinstatement on RT. (C-F) Frontoparietal ROIs showing significant reinstatement of CTD. From left to right: locations of ROI (marked in red), group mean (\pm MSE) of context-trial pattern similarity plotted as a function of experimental conditions, quintiles (x-axis) of group mean of log-transformed RT (\pm MSE) plotted as a function of task type and context-trial pattern similarity. The names of the ROIs are from the Human Connectome Project's multi-modal cortical parcellation⁹. *: $p < 0.05$; **: $p < 0.01$; ***: $p < 0.001$. con: congruent, inc: incongruent.

Figure 5. Hippocampal activity and pattern separation modulate cortical reinstatement of CTD. (A) Four examples (grouped by blue background) of how the CTDs of context A (top row) and context B (bottom row) define the experimental condition of context-context pattern similarity (text label below the background). (B) Visualization of the hippocampus ROI (top, in red) and the group mean (\pm MSE) of the three conditions of hippocampal context-context pattern similarity (bottom). (C) Group mean (\pm MSE) of the modulation index of each of the four predictors and the ROI univariate activity on the CTD reinstatement in each of the four frontoparietal ROIs. (D) Quintiles (x-axis) of group mean of CTD reinstatement in BA 9-46d (\pm MSE) plotted as a function of hippocampal

context-context pattern similarity in the Same CTD condition (left panel) and hippocampal univariate activity (right panel). *: $p < 0.05$; **: $p < 0.01$.

****REVIEWERS' COMMENTS:**

Reviewer #1 (Remarks to the Author):

I appreciate the authors have improved the manuscript extensively with additional analyses to clarify and strengthen the arguments. I have only a few minor points for further polishing.

1. (Regarding Point 6 and the response) The model in Fig. 1 proposes that context-CTD association promotes pattern separations between the contexts which share the same CTD to prevent the context-context interference from the context1-CTD-context2 association. Because a stronger context-CTD association would be positively related to stronger CTD retrieval, stronger pattern separation would also lead to better CTD reinstatement. (If there is another dot in Fig. 1B for the CTD associated to the two contexts, the context-CTD association is stronger for Strong separation than Weak separation.) However, the authors hypothesized that Weaker context-context separation may facilitate CTD retrieval through recurrent process across context1-CTD-context2 association loop (related to Fig. 1B and Fig. 5D). There is a gap in the argument as to why the association can be either interruptive or facilitative in the first place. The pattern separation of two same-CTD contexts may also cause the separation between the same CTD if the CTD retrieval includes related contextual reinstatement (note: It would be interesting to see if stronger context-context pattern separation predicts the CTD in the specific context). Weak separation may indicate more integration across context1-CTD-context2 association rather than competition, which could support the authors' prediction. Without more sound logic, this hypothesis seems too post-hoc. This argument may be more suitable in the discussion section rather than introduction as a hypothesis.

2. I appreciate that the authors provided individual data in Supplementary Fig. 3 & 4. However, it would be more digestible if means and standard errors are plotted as well.

Reviewer #2 (Remarks to the Author):

I am overall satisfied with the authors' responses, but for one:

#2a. The authors appear to agree that the directionality claims are unsupported by their data, but do not remove them from the manuscript - in the abstract, Figure 1, etc. You have enough results in here otherwise, it's not clear that you need to assert directionality beyond the ability to resolve such.

Reviewer #3 (Remarks to the Author):

The revision has clarified many of the issues in the previous submission. I have a few minor suggestions that relate to wording:

- "Distinct from the pattern separation theory that different memory traces are encoded in sparse neural codes, this effect assumes partially overlapping neural codes for overlapping experiences." It does not make sense to draw a parallel between a theory and an experimental effect. It would make more sense to compare how the data relates to the theory. In this case, the authors would need to explicitly state that the data are not compatible with the theory, or if they believe otherwise, try to reconcile the data with the theory. It is also odd that the Nonmonotonic plasticity hypothesis is mentioned but not explicitly compared with the pattern separation account. I think the authors need to take a stand as to where the theories line up, or to say that there are other possibilities, it isn't really enough to describe the data.

- "direct modulation from the hippocampus to behavior". It isn't clear that the hippocampus modulates behavior, rather that there is a direct relationship between hippocampal PS and behavior (i.e., it is behaviorally relevant).

- line 324: "effort" should be "effect".

- "we leveraged computer-graphics techniques to create spatial contexts" It's sufficient to say that subjects performed the task in VR environments.

- line 522: "We adopted an incidental learning paradigm, as this is typical in the literature of item-specific learning of cognitive control demand." This wording is awkward but also not really descriptive of the study. The authors themselves describe the task in terms of contextual task demands, so this isn't about item-specific learning.

- Lines 524-526: "explicitly learned CTD will result in similar retrieval processes, given the importance of hippocampus in the retrieval of explicitly formed associations. For example, the hippocampal pattern differentiation findings (c.f., Fig. 5B) were first observed in studies exploring explicit associations..."

-> The authors appropriately state that their task did not necessarily involve implicit learning, but then this statement seems to contradict that assertion. I think it is useful to distinguish between incidental vs intentional learning (which is relevant to the current study) as compared to implicit vs explicit learning (which is irrelevant to the design or points made here). Unfortunately, many readers (including specialists) don't seem to understand that difference, and I think it's important to clarify the point if the authors are going to include this comment.

Charan Ranganath (I sign all reviews)

Response to reviews

We thank Dr. Ranganath and the two anonymous reviewers for their positive feedback and thoughtful comments, which will further strengthen the manuscript. We revised the manuscript accordingly, as described in the below point-by-point responses to the comments.

Reviewer #1

1). (Regarding Point 6 and the response) The model in Fig. 1 proposes that context-CTD association promotes pattern separations between the contexts which share the same CTD to prevent the context-context interference from the context1-CTD-context2 association. Because a stronger context-CTD association would be positively related to stronger CTD retrieval, stronger pattern separation would also lead to better CTD reinstatement. (If there is another dot in Fig. 1B for the CTD associated to the two contexts, the context-CTD association is stronger for Strong separation than Weak separation.) However, the authors hypothesized that Weaker context-context separation may facilitate CTD retrieval through recurrent process across context1-CTD-context2 association loop (related to Fig. 1B and Fig. 5D). There is a gap in the argument as to why the association can be either interruptive or facilitative in the first place. The pattern separation of two same-CTD contexts may also cause the separation between the same CTD if the CTD retrieval includes related contextual reinstatement (note: It would be interesting to see if stronger context-context pattern separation predicts the CTD in the specific context). Weak separation may indicate more integration across context1-CTD-context2 association rather than competition, which could support the authors' prediction. Without more sound logic, this hypothesis seems too post-hoc. This argument may be more suitable in the discussion section rather than introduction as a hypothesis.

Response: We thank the reviewer for highlighting the need for further clarity here. Following the reviewer's suggestion, we simplified the Introduction by moving the argument related to the negative effects of context-context separation on CTD reinstatement to the Discussion (see below). Concurrently, we modified Fig. 1 to better emphasize the hypothesis that when two contexts have overlapping features (i.e., similar CTDs), theory predicts that their representations in hippocampus should be pattern separated/differentiated. Here is the modified Fig. 1:

Figure 1. Proposed neurocognitive processes underlying the reinstatement of learned CTD. Blue text indicates the figures where relevant results are shown.

In our previous response, we intended to communicate that strong pattern separation can be disruptive or facilitative *depending on the goal*. For the goal of distinguishing between two contexts, weak separation is always detrimental as it introduces more interference. Importantly, and by contrast, in the case of retrieving the associated CTD, weak separation may be facilitative (either through integration or inference at retrieval). As noted above, we moved this argument to the Discussion and revised text for clarity:

“In the present paradigm, we hypothesized that context-level differentiation may support achieving the goal of representing the particular context in which one is situated as well as the goal of remembering the probabilistically dominant task performed within the context. While a priori one might predict that hippocampal pattern separation would similarly benefit both goals, it is possible that the effects can differ. On the one hand, if the goal is to distinguish between two contexts, hippocampal coding of contexts within each pair may be separated to counter the interference between contexts caused by the shared CTD. On the other hand, pattern separation may be disruptive if the goal is to retrieve the associated CTD: Specifically, consider two pairs of contexts (dark and light dots in Fig. 6, each dot representing one context), each associated with one CTD. Relative to weaker hippocampal pattern separation (right panel of Fig. 6), when pattern separation is strong (left panel of Fig. 6) the cuing of one context is less likely to concurrently retrieve, or suffer interference from, the other context sharing the same CTD. However, while such strong pattern separation might keep the contexts more distinct, this may result in a failure to leverage the other context to boost retrieval and reinstatement of the shared CTD through recurrent retrieval^{1, 2}. While speculative, the divergent effects on the two goals may explain why overlapping contexts and their shared CTD were, on average, not linked in a single memory trace through integrative encoding³ (which would have been evidenced by increased pattern similarity between contexts in the Same CTD condition). At the same time, we observed that dlPFC reinstatement of CTD positively scaled with the hippocampal pattern similarity between the two overlapping contexts (i.e., Same CTD condition, see Fig 5C-D). This observation is broadly consistent with models of hippocampal generalization^{1, 2}, as weaker pattern separation may allow recurrent retrieval of the other context sharing the CTD, which would facilitate the retrieval and reinstatement of the associated CTD. Future research should explore whether and how recurrent or chained retrieval of associated task-sets and sensory information support prospective planning and flexible behavior in complex tasks⁴.”

Figure 6. Relationship between pattern separation of context representations and CTD retrieval. Stronger hippocampal separation of contexts sharing the same CTD may lead to more distinct neural coding of contexts (indicated by the distance between dots of the same color), but weaker facilitation from the other context in supporting retrieval of the associated CTD (indicated by the thickness of the red arrows).

Finally, we added clarifying language in two locations related to the non-monotonic hypothesis:

- a) We further clarify that within two partially overlapping associations, the relationship between association strength and the degree of separation is non-monotonic based on a recent theory (Ritvo et al., 2019): "... strong separation does not necessarily result from strong association strength: the relationship between the two can be non-monotonic⁵."
- b) To account for the dynamics of pattern separation (e.g., separation can change after each retrieval), context-context pattern similarity was calculated at the block level (i.e., one similarity score for each context onset) when testing the relationship between hippocampal pattern separation and PFC reinstatement of CTD: "To account for the possible change in pattern separation after each encounter of a context, context-context pattern similarity was calculated at the onset of the room for each block, again for Same Context, Same CTD, and Different CTD conditions (Fig. 5A)"

2) I appreciate that the authors provided individual data in Supplementary Fig. 3 & 4. However, it would be more digestible if means and standard errors are plotted as well.

Response: Following the reviewer's suggestion, we also now plot means and standard errors in Supplementary Fig. 3 & 4. These revised figures are also presented here for ease of consideration:

Supplementary Figure 3. Individual (left panel) and mean \pm SEM of (right panel) context-context pattern similarity for Same Context and Different CTD conditions. Each line represents one participant.

Supplementary Figure 4. Individual (upper panels) and mean \pm SEM of (lower panels) context-context pattern similarity, plotted as a function of experimental condition (Same Context and Different CTD) and time (run 1 and runs 2-6). Each line represents one participant.

Reviewer #2

1) *The authors appear to agree that the directionality claims are unsupported by their data, but do not remove them from the manuscript - in the abstract, Figure 1, etc. You have enough results in here otherwise, it's not clear that you need to assert directionality beyond the ability to resolve such.*

Response: We thank the reviewer for noting the richness of the data we report, as well as for seeking further consideration of this issue. In our response to the previous round of review, we provided additional findings supporting the directionality of hippocampal modulation on CTD reinstatement in dlPFC (Supplementary Note 7). While not definitive, we believe the data lend support for this aspect of our directionality claims, which are also consistent with theories of hippocampal mechanisms being the drivers of event feature reinstatement in cortex. To further address this point, we modified the text to the following:

a) To temper the prior causal language, when describing the findings in Fig. 5 (in the manuscript title, abstract, and main text), we replaced the words “mediate” and “modulate” with either “predict”, “correlate” or “co-vary”.

b) We further note that: “Collectively, these results are consistent with theories of pattern completion that posit that the hippocampus drives restatement of associated event features (here the CTD) in cortex (here dlPFC/frontopolar). While fMRI data lack the temporal resolution to definitively demonstrate that the hippocampal effect precedes the CTD reinstatement in PFC, we performed additional control analyses to discount the possibility that the observed hippocampal-PFC relationship was due to a modulation from dlPFC/frontopolar cortex to the hippocampus or to a common modulator (Supplementary Note 7). Again, definitive evidence on this point awaits further experimentation.”

Please note that we also changed Fig. 1 in response to a comment from R1. With respect to the prediction in Fig. 1 regarding the temporal relationship between CTD reinstatement and trial-wise RTs, we note that this directly stems from the task design. Specifically, the former occurred at the onset of the context (i.e., at the beginning of each block), which was earlier than the trials that appeared within the block.

Reviewer #3

1) *"Distinct from the pattern separation theory that different memory traces are encoded in sparse neural codes, this effect assumes partially overlapping neural codes for overlapping experiences." It does not make sense to draw a parallel between a theory and an experimental effect. It would make more sense to compare how the data relates to the theory. In this case, the authors would need to explicitly state that the data are not compatible with the theory, or if they believe otherwise, try to reconcile the data with the theory. It is also odd that the Nonmonotonic plasticity hypothesis is mentioned but not explicitly compared with the pattern separation account. I think the authors to need to take a stand as to where the theories line up, or to say that there are other possibilities, it isn't really enough to describe the data.*

Response: Following the reviewer's suggestion, we revised the text to discuss this finding in relation to pattern separation theory and the nonmonotonic plasticity hypothesis. The text now reads:

"As with the preceding studies, this finding might to be at odds with pattern separation theory, which posits that pattern separation drives the hippocampal representations of overlapping events to be distinct as would be expected by default for the representations of non-overlapping events. However, such hyper pattern distinctiveness for overlapping events may be explained as a secondary effect that follows initial pattern separation. Specifically, assuming that pattern separation generates initial orthogonal representations for events sharing the same feature, the nonmonotonic plasticity hypothesis proposes that pattern similarity between the events can decrease further through a pruning process that follows weak activation of the overlapping portion of the neural codes⁵. Indeed, analyses of the temporal profile of change in pattern similarity in the Same CTD condition lends support for this latter interpretation, as the hyper pattern distinctiveness (i.e., lower similarity in Same CTD than in Different CTD) emerged over time (Supplementary Note 6)."

2) *"direct modulation from the hippocampus to behavior". It isn't clear that the hippocampus modulates behavior, rather that there is a direct relationship between hippocampal PS and behavior (i.e., it is behaviorally relevant).*

Response: We thank the reviewer for this comment. Following the reviewer's guidance, we changed the text to read: "The behavioral relevance of the present hippocampal pattern distinctiveness effect is supported by the observed relationship...".

3) *line 324: "effort" should be "effect".*

Response: Thank you; fixed.

4) *"we leveraged computer-graphics techniques to create spatial contexts" It's sufficient to say that subjects performed the task in VR environments.*

Response: The sentence now reads: "participants performed the task in a 3D virtual environment with spatial contexts (i.e., buildings and rooms)."

5) *line 522: "We adopted an incidental learning paradigm, as this is typical in the literature of item-specific learning of cognitive control demand." This wording is awkward but also not really descriptive of the study. The authors themselves describe the task in terms of contextual task demands, so this isn't about item-specific learning.*

Response: We appreciate the Reviewer highlighting the potential for confusion here. Our intent is to communicate that the experimental design was inspired by previous studies investigating item-specific learning of cognitive control demand. We changed the wording accordingly to:

“Inspired by the literature of item-cognitive control demand associations, we adopted an incidental learning paradigm.”

6) Lines 524-526: "explicitly learned CTD will result in similar retrieval processes, given the importance of hippocampus in the retrieval of explicitly formed associations. For example, the hippocampal pattern differentiation findings (c.f., Fig. 5B) were first observed in studies exploring explicit associations..." The authors appropriately state that their task did not necessarily involve implicit learning, but then this statement seems to contradict that assertion. I think it is useful to distinguish between incidental vs intentional learning (which is relevant to the current study) as compared to implicit vs explicit learning (which is irrelevant to the design or points made here). Unfortunately, many readers (including specialists) don't seem to understand that difference, and I think it's important to clarify the point if the authors are going to include this comment.

Response: We thank the reviewer for raising this important conceptual issue. The flagged language was added to address Reviewer 1's previous comment on explicit learning, and thus we believe it is important to retain. To stress the difference between learning type (explicit vs. implicit) and experimental manipulation (intentional vs. incidental), we added the following clarification:

“We expect that explicitly learned CTD (which can result from intentional or incidental learning) will result in similar retrieval processes, ...”

References

1. Kumaran, D. & McClelland, J.L. Generalization through the recurrent interaction of episodic memories: a model of the hippocampal system. *Psychological review* **119**, 573-616 (2012).
2. Koster, R., *et al.* Big-Loop Recurrence within the Hippocampal System Supports Integration of Information across Episodes. *Neuron* **99**, 1342-1354 e1346 (2018).
3. Shohamy, D. & Wagner, A.D. Integrating memories in the human brain: hippocampal-midbrain encoding of overlapping events. *Neuron* **60**, 378-389 (2008).
4. Brown, T.I., *et al.* Prospective representation of navigational goals in the human hippocampus. *Science* **352**, 1323-1326 (2016).
5. Ritvo, V.J.H., Turk-Browne, N.B. & Norman, K.A. Nonmonotonic Plasticity: How Memory Retrieval Drives Learning. *Trends in cognitive sciences* **23**, 726-742 (2019).